# TP53-dependent toxicity of CRISPR/Cas9 cuts is differential across genomic loci and can confound genetic screening

Miguel M. Álvarez [1], Josep Biayna [1,3] & Fran Supek [1,2] ✉

CRISPR/Cas9 gene editing can inactivate genes in a precise manner. This process involves DNA double-strand breaks (DSB), which may incur a loss of cell fitness. We hypothesize that DSB toxicity may be variable depending on the chromatin environment in the targeted locus. Here, by analyzing isogenic cell line pair CRISPR experiments jointly with previous screening data from across ~900 cell lines, we show that *TP53*-associated break toxicity is higher in genomic regions that harbor active chromatin, such as gene regulatory elements or transcription elongation histone marks. DSB repair pathway choice and DNA sequence context also associate with toxicity. We also show that, due to noise introduced by differential toxicity of sgRNA-targeted sites, the power of genetic screens to detect conditional essentiality is reduced in *TP53* wild-type cells. Understanding the determinants of Cas9 cut toxicity will help improve design of CRISPR reagents to avoid incidental selection of *TP53*-deficient and/or DNA repair deficient cells.

The widespread adoption of CRISPR/Cas9 gene editing technology[1–4] has revolutionized the systematic study of gene essentiality in mammalian cells[5]. The study of genetic interactions and gene-drug associations[6] using Cas9 gene editing is especially active in the field of cancer research, allowing the identification of genes essential within the genetic context of a tumoral tissue but not in healthy tissue, or within a particular genetic background (commonly involving a mutated tumor suppressor gene), or the identification of genes whose inactivation in a tumor boosts the therapeutic effect of a drug[7–10].

Application of the CRISPR/Cas9 technology to perform gene knockout (KO) is based on introducing a double-strand break (DSB) into a coding region of a gene of interest via the Cas9 endonuclease bound to a single-guide RNA (sgRNA), a part of which is complementary to the target site. Frameshifting indels can be introduced in the process of DSB repair, inactivating the gene[11], more reliably so if the frameshift also triggers the nonsense-mediated decay pathway to degrade the mRNA[12]. The CRISPR/Cas9 system is often used for genome-wide genetic screening experiments, where the process usually begins with the lentiviral transduction of a cell culture with a pool of sgRNAs that target virtually all known human genes, aiming to introduce one sgRNA into each cell. Then, one can systematically estimate gene essentialities by comparing sgRNA counts (obtained through DNA sequencing) between a pool of cells sampled at an early time-point, and a pool of cells sampled at the end of the experiment, after the gene KOs have exerted effects on cell fitness. By extension, conditional gene essentiality (i.e., that specific to a treatment) can be estimated by comparing sgRNA gene abundance in a control versus a treated pool of cells.

The essentiality of a gene may depend on the genetic background of the studied cell line[5]. One special case of this is the *TP53* functional status of a cell, where *TP53*-mutant cells are less likely to arrest or undergo apoptosis due to DSBs, including those resulting from Cas9[13–16]. Recent reports have pointed out difficulties in capturing signals of gene essentiality in CRISPR/Cas9 KO screening data when using *TP53* wild-type cell lines[14,15], presumably due to greater toxicity of DSBs[13]. However it is not clear if these results hold true universally[17,18].

[1]Genome Data Science, Institute for Research in Biomedicine (IRB Barcelona), Barcelona institute for Science and Technology, Barcelona, Spain. [2]Catalan Institution for Research and Advanced Studies (ICREA), Barcelona, Spain. [3]Present address: Department of General, Visceral, Transplant, Vascular and Pediatric Surgery (Department of Surgery I), University Hospital Würzburg, Würzburg, Germany. ✉e-mail: fran.supek@irbbarcelona.org

Motivated by reports of DSB toxicity interfering with gene essentiality readouts in CRISPR/Cas9 screens, we were interested in the impact of p53 status in cell line screens where gene essentiality is compared between two or more conditions. These experiments may be used, for instance, to reveal synthetic lethality relationships. We studied this on an isogenic pair of cell lines that are wild-type and KO for *TP53*, derived from human lung adenocarcinoma A549 cells. The generation and assaying of *TP53*-isogenic cell lines has been done previously for RPE1[14,17–19] and MOLM13[20] cell lines. The treatment here consisted of a chemical inhibition of the *ATR* protein (ATRi), for which many conditionally essential genes are known[21]. Interestingly, we detected *TP53*-dependent effects on cell fitness at the individual sgRNA-level, rather than gene-level, suggesting this was not due to disruption of gene function as intended by the screening. We further investigated the effect of this differential sgRNA toxicity on the power to identify gene-ATRi interactions. A systematic, genome-wide analysis of toxicities identified multiple local correlates of this p53-associated effect, such as active chromatin marks and proximity to certain oligonucleotide motifs, thus suggesting avenues for selecting gene editing sites to avoid toxicities to *TP53* wild-type cells. This knowledge could also be applied to another promising application of CRISPR, therapeutic in vivo or ex vivo gene editing, where potential for selection of *TP53*$^{-/-}$ cells is a concern[13,20,22] that may be allayed by a judicious choice of loci targeted for editing.

## Results

### p53-mediated DSB toxicity does not prevent identifying universally essential genes

Following recent reports that presented apparently divergent data[14,15,17,18], we first asked whether *TP53* status affects the ability to recover universally essential genes in a CRISPR/Cas9 genetic screen. We thus examined our A549 *TP53* wild-type (TP53wt) and knockout (TP53$^{-/-}$) isogenic pair for ability to correctly classify known essential[11] from non-essential[23] genes, considered as a set (see Methods; we note that this benchmark considers two sets of genes on the extremes of the fitness effect spectrum, and so it may not be representative of the genes with more subtle fitness effects). We used three biological pseudo-replicates, which differ in their treatment: either untreated, doxycycline-treated, or doxycycline and ATRi-treated (see Methods for details). The areas under the ROC curve (AUROC) for the later time points (Supplementary Table 1) indicate that the sgRNA dropout patterns accurately describe the essentiality or non-essentiality of a gene, irrespective of the *TP53* status. The high AUROC values, as well as the TP53wt data not showing worse accuracy than TP53$^{-/-}$, go in line with another recent study that considered *TP53*-isogenic pairs[17] (Supplementary Table 2). Next, we examined sgRNA depletion levels. These correlated with the essentiality of the targeted gene regardless of *TP53* status (Fig. 1a). However, the counts of non-targeting sgRNAs (which do not have a sequence match to any genomic region) are systematically higher than counts of sgRNAs targeting non-essential genes, even in TP53$^{-/-}$ samples, suggesting a general toxicity of Cas9 DSBs or downstream events. Next, we asked whether this toxicity is different between *TP53* backgrounds. Remarkably, while there was no difference between TP53wt and TP53$^{-/-}$ samples regarding the sgRNA counts of non-essential genes (Supplementary Fig. 1), TP53wt samples had higher counts of the non-targeting sgRNAs. This suggests that p53 activity exacerbates DSB toxicity, such that a lack of DSB is particularly advantageous to TP53wt cells, in relative terms. Overall, our data suggests that DSB created by Cas9 in various genomic locations can be toxic via a *TP53*-dependent mechanism, and also supports previous claims[17,18] that *TP53* status only modestly affects the ability to identify the universally essential genes, considered as a set, in human cultured cells.

### TP53 wild-type background can confound estimates of gene selection

We were further interested in whether the p53-mediated fitness loss can confound CRISPR/Cas9 essentiality assays, with respect to the power to identify individual essential genes. We performed analyses using MAGeCK-MLE[24] on the A549 cell line, at 18 data points (see Methods). Our aim was to characterize the proportions of genes whose sgRNAs are either depleted (significantly negative beta score of MAGeCK-MLE, i.e., essential genes), or increased (significantly positive beta score, i.e., potential tumor suppressor genes) at time points t9, t12, and t15, with respect to time point t0 (Fig. 1b). Across all nine conditions, TP53wt samples have more genes showing a significant signal of negative selection than TP53$^{-/-}$ samples do (Mann-Whitney $p = 0.00004$). This holds also when excluding the ATRi-treated samples (Mann-Whitney $p = 0.00216$ for both negative and for positive selection), suggesting that the TP53wt background inflates estimates of selection for some genes also under non-stressed conditions.

In light of this, we asked whether the inflated amount of apparently selected genes in TP53wt could be explained solely by the differential selection of sgRNAs that target genes related to the p53 pathway, which might plausibly be epistatic with *TP53*. Alternatively, these apparently selected genes may be spurious hits caused by p53-mediated DSB toxicity, which would be unrelated with the function of the gene targeted by the sgRNA. We identified genes systematically negatively selected exclusively in (*a*) TP53wt samples (*n* = 61), and (*b*) TP53$^{-/-}$ samples (*n* = 6 genes) (Supplementary Table 3). Of the 61 genes, only four (*MDM2*, *MDM4*, *USP7*, and *AURKA*) are among the top-50 *TP53* interacting genes (gene functional associations as per STRING database[25]; see Methods). *MDM2* and *MDM4* are known negative regulators of p53 activity, therefore it is expected that loss of *MDM2* and *MDM4* would activate p53, leading to growth arrest or apoptosis of TP53wt cells, however not of TP53$^{-/-}$ cells[26]. The remaining 57 genes may be artifactual hits, identified due to DSB cut toxicity at particular loci but not gene function.

### Guide-level effects rather than gene-level effects may underlie the p53-associated fitness loss

To investigate, we consulted conditional essentiality data from 649 *TP53*-mutant (TP53mut) and 257 TP53wt cell lines (classification detailed in Methods) from Project Achilles[27] and PScore[28] combined[29], pooled across tumor types. The negatively selected genes that overlap with the 61 genes set from the A549 analyses are, again, *MDM2*, *MDM4*, and *USP7*, as well as only nine other genes (see Methods). To additionally support that the majority of the 61 hits are likely not epistatic with *TP53*, we also ran a Gene Ontology (GO) enrichment analysis[30], where the enriched GO terms related to DNA damage response or to cell cycle regulation account for only 23 of the 61-set genes (including among others *MDM2*, *MDM4*, *USP7*, and *AURKA*, see Supplementary Table 4). Figure 1c summarizes the gene overlaps described here. In summary, the majority of the genes under apparent negative selection in TP53wt cells do not appear related by function to *TP53*. This suggests that something other than the function of the targeted gene — such as differential p53-mediated DSB toxicity across various loci in the genome— causes this readout.

Next, we additionally analysed the overlap of genes negatively selected exclusively in TP53wt samples between various genome-wide CRISPR screening libraries, comparing our A549 data with previous data from the RPE1 cell line (normal retinal pigment epithelium) *TP53*-isogenic pairs, which used Brunello[14,31] and Gecko v2[19,32] libraries. The concordance of *TP53*-associated hits between experiments using the same library but in different cell lines is, interestingly, similar or larger than the concordance between different libraries on the same cell line (see Supplementary Text 1a and Fig. 1d). This is compatible with differential p53-mediated DSB toxicity underlying our observations: because the sgRNAs of different libraries target different loci within

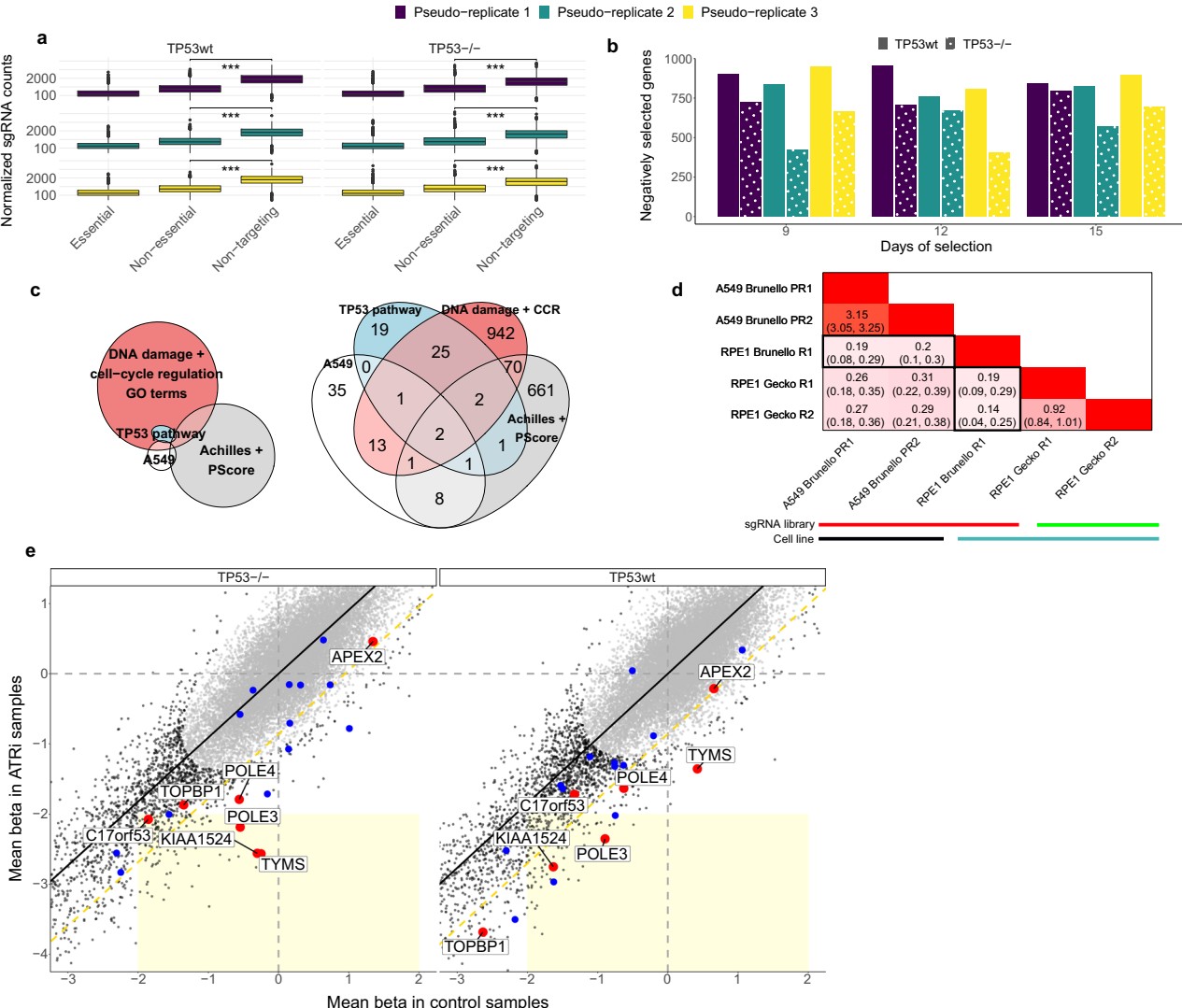

**Fig. 1 | A *TP53* wild-type background can confound estimates of gene selection in genetic screens. a** Boxplots showing the pooled normalized sgRNA counts per sample (essential and non-essential genes, and non-targeting sgRNAs; 15 day samples are shown). Tested using 1-tailed Mann-Whitney. *** denotes a *p* <2.2e-16. No adjustments were made for multiple comparisons. *n* = 7300 independent sgRNAs examined over six independent experiments. **b** Barplot showing the number of genes that are negatively (beta score<0) selected, per sample used in this study. Beta score significance: FDR < 0.25. **c** Venn (left) and corresponding Euler (right) diagrams of the overlap of genes between four sets: genes negatively selected exclusively in TP53wt in our samples (A549), genes negatively selected exclusively in TP53wt in Project Achilles and Score (Achilles + Score), top-50 TP53-interactors (TP53 pathway), and genes included in 19 GO terms related to DNA damage and cell-cycle regulation that we found enriched with genes from the A549 set. **d** Results of the analysis of overlap between different cell lines and/or sgRNA libraries detailed in Supplementary Text 1a: heatmap shows the log$_2$ odds ratio of the overlap of genes negatively selected exclusively in TP53wt, between different

experiments. R: Replicate, PR: Pseudo-replicate. Darker shades of red indicate higher overlap. Black rectangles highlight the overlap between RPE1 Brunello dataset with others. **e** Comparison between *TP53*-isogenic cell lines to assess biases in identifying conditional essentiality from genetic screens. *x* and *y* axes represent the standardized beta scores (*Z*-scores) for genes either in the control samples (incl. doxycycline-treated; pseudo-replicates 1 and 2), or in the doxycycline+ATRi treated samples, respectively, averaged across later time points and pseudo-replicates. Coordinate axes were capped in order to zoom on the region of interest. The EM clustering identified two gene clusters as the most likely model, represented by black and gray dots. Black line represents the best fit linear model. The yellow dashed diagonal line represents −2 standard deviations (SD) of the *Z*-score difference. The light yellow rectangle delimits the tentative significance area containing genes negatively selected in the treatment, but not selected in the control sample (i.e., potentially synthetic lethal with ATRi). The top-20 validated ATRi-sensitizing genes are highlighted with color, and the top-7 (red) are further labelled. Source data are provided as a Source Data file.

the same gene, they may elicit different DSB toxicity and thus fitness effects, even though their effects on the function of the targeted gene are presumably similar.

### *TP53* status can bias genetic screens for conditional essentiality

The analyses below suggest that thousands of sgRNAs from a genome-wide CRISPR screening library can be subject to *TP53*-dependent negative selection, in a manner largely unrelated to the function of the targeted gene but instead associated with another property of the

locus. The high number of toxic loci (3308) contrasts with the much lower number of genes (61) detected as significantly differentially selected between TP53$^{-/-}$ and wild-type A549 cells (possibly because MAGeCK-MLE −used to define the 61 gene set− downweights outlier sgRNAs, so the effect of p53-toxicity in one sgRNA per gene could be partially adjusted). Because usually at most one sgRNA per gene (out of four per gene in this library) is *TP53*-conditionally toxic (see Supplementary Fig. 2), pooling fitness data per gene may be able to overcome the noise this variable toxicity introduces. Thus a benchmark set of

known essential genes can be accurately identified in our A549 iso-genic pair experiment, irrespective of the *TP53* background (see Supplementary Table 1 and Fig. 1a), consistent with similar recent analyses on the RPE1 cell line *TP53*-isogenic pair[17,18]. This benchmark may be considered easy because this set of essential genes is known to have a fitness effect in cell culture irrespective of genetic background, and also because their fitness effect is usually strong (see Supplementary Fig. 3a). Nevertheless, many additional essential genes may exist, whose fitness effects are more subtle and/or conditional on the genetic background, or, crucially, conditional on treatment conditions (see Supplementary Fig. 3b).

We asked if the *TP53* status can confound estimates in such cases when conditional selection is measured by comparing a treatment and control arm of an experiment —a common usage scenario for genetic screens. Because Cas9 targeting of many non-essential genes triggers p53-dependent toxicity in A549 cells (see above), these genes could result in false negative errors in a comparison of two conditions, such as presence versus absence of a drug. This is because they would be depleted in both conditions due to the cut toxicity, thus the observed effect sizes may be attenuated. Further, sgRNAs targeting different genes and also different loci in the same gene may have varied fitness effects in part due to the locus-specific toxicity of the cut. This would introduce noise, making it harder to detect statistically significant differences between experimental conditions.

Overall, we hypothesized that the signal-to-noise ratio to detect conditionally essential genes would be less favorable in a TP53wt background compared to a TP53⁻/⁻ background. To examine this, we compared gene essentialities between the combined untreated and doxycycline-treated pseudo-replicates (see Methods) versus the pseudo-replicates treated with an *ATR* inhibitor drug (ATRi, see Methods). As a benchmark, we used known ATRi-sensitizing genes validated previously by compiling results across multiple human cell lines[21] (of note, these did not include A549 cells). We considered a stringent set of validated ATRi-sensitizing genes (top-7, see Methods) that are almost certain to be hits in any genetic background. We used MAGeCK-MLE to estimate selection in the ATRi-treated and the control A549 samples (Fig. 1e), identifying two clusters of selection coefficients across the TP53wt and TP53⁻/⁻ cells, which broadly correspond to non-selected genes and negatively selected genes. This latter cluster contains six of the top-7 genes (*KIAA1524/CIP2A*, DNA polymerase ε accessory subunits *POLE3* and *POLE4*, *TYMS*, *C17orf53/HROB*, and *TOPBP1*) in both *TP53* backgrounds, indicating that our experiments generally mirror the set of ATRi-sensitizing genes identified previously. Next, we examined ATRi conditional essentiality in our data by considering standardized MAGeCK-MLE gene selection coefficients (*Z*-scores; Methods), tested separately for the treatment condition (here, ATRi) and the untreated condition. Across both *TP53* backgrounds, in the top-7 set only the genes *KIAA1524/CIP2A* and *POLE3* would pass this test for conditional selection (i.e. negative selection in the treatment, and no evidence of negative selection in the control). Additionally, the TP53⁻/⁻ background —but not the TP53wt— would identify *TYMS* and *C17orf53/HROB*. Conversely, there are no top-7 genes that are identified only in the TP53wt background. The known *POLE4* gene is a near-hit with a stronger signal in the TP53⁻/⁻ background (*Z*-score = −1.79) than in the TP53wt background (*Z*-score = −1.63). Overall, this suggests that the genes which sensitize to a drug treatment are more readily recovered in the TP53⁻/⁻ background than in the TP53wt background of the same cell line. To refine this analysis, we examined the residuals of the fit between *Z*-scores from ATRi condition and controls in the TP53wt background, compared with the residuals of this same fit in the TP53⁻/⁻ background (Supplementary Fig. 4a). In four of the known top-7 genes, the residuals are more negative (i.e., higher degree of ATRi-conditionality) in the TP53⁻/⁻ background (*TYMS*, *KIAA1524/CIP2A*, *POLE3*, *POLE4*), while they are more negative in TP53wt in only two genes (*C17orf53/HROB* and *TOPBP1*). This above suggests that the

power to detect gene conditional essentiality is hampered in TP53wt cell lines compared to their TP53⁻/⁻ counterpart.

Next, we checked that our results are not specific to one statistical method, employing two different software: drugZ[33] and BAGEL v2[34] (see Methods). The results support the hypothesis of a lower power of detection of conditional essentiality in TP53wt cell lines (Supplementary Fig. 4b, c). We highlight an example gene, *KIAA1524/CIP2A*, with one discordant sgRNA within the gene showing *TP53*-associated effects (Supplementary Text 1b). In addition, we ran a complementary analysis that suggests that the TP53wt background can impede discovery of unreported synthetic lethal genes (here, *FBXW7*, see Supplementary Text 1c).

### sgRNA-level analyses of cut toxicity reveal genomic and epigenomic determinants

We asked what are the properties of sgRNA-targeted loci that result in stronger toxicity. Because pooling data across multiple sgRNA loci in a gene could obscure differences between loci, we focused on the sgRNA-level (instead of gene-level) negative selection that is *TP53*-dependent (see Methods). We identified 3308 sgRNAs that were negatively selected in TP53wt relative to TP53⁻/⁻ A549 cells (henceforth target loci, see Methods). The total number of genes in which at least one sgRNA locus is affected by *TP53*-dependent negative selection is 2990, reflecting that usually only one sgRNA was negatively selected within a gene (see Supplementary Fig. 2). In addition, we note 2559 sgRNAs that were apparently positively selected in TP53wt relative to TP53⁻/⁻ A549 cells; an analysis of LFC mean-to-variance ratio for sgRNAs across the three pseudo-replicates suggests that this set may be enriched with false positives (Supplementary Fig. 5a), and also that part of the positive selection may be due to targeting some of the top-50 genes with functional interactions with *TP53* (STRING; O.R. = 2.78, 95% C.I. = [1e-3, 7885]) rather than sgRNA level effects.

Out of the 33 genes that we identified to have a differential p53-related toxicity between sgRNA target positions (see *TP53 wild-type background can confound estimates of gene selection* above), only seven of them were identified to contain (as maximum) one target locus. This is largely due to different stringency thresholds between two algorithms employed for the two analyses (see Methods). A key difference is that the gene-level MAGeCK-MLE analysis downweights outlier sgRNAs, so the effect of p53-toxicity in one sgRNA per gene could be partially adjusted. Furthermore, the mean sgRNA LFC values are lower among the 33 genes than in the remaining library (Mann-Whitney *p* < 2.2e-16), supporting that the gene-level and guide-level analyses are not contradictory, but that the mean LFC threshold employed to select the p53-toxic sgRNAs was very strict, moreso than the MAGeCK-MLE analysis applied to the gene-level analysis.

We also defined another set of sgRNAs: those belonging to the same genes as the target loci but not exhibiting *TP53*-dependent toxicity, henceforth background loci, thus controlling for possible effects on gene function. Using these two sets of loci, we investigated the DNA sequence and epigenomic determinants that associate with p53-mediated toxicity. As a control for the expected off-target effects we also defined a set of 4,027 confidently non-selected sgRNAs (henceforth non-selected loci, see Methods). This suggested that sgRNA off-targeting does contribute somewhat to p53-associated toxicity, however also that off-targeting explains only a minor part (-13%) of the p53-toxic loci (Supplementary Text 2a), which were excluded from subsequent analysis.

### Association of active chromatin features with high-p53-toxicity sgRNA target sites

We further hypothesized that the toxicity of Cas9-induced DSBs may be modulated by the chromatin state of the break locus, since diverse DSB repair mechanisms are known to be associated with heterochromatin, lamina-associated domains (LAD), and certain histone

modifications[35–37]. We additionally considered chromatin accessibility (as estimated by the DNase-I hypersensitive sites (DHS)) DNA replication timing (RT), and gene transcription levels (estimated by mRNA abundance), since these factors were associated with mutation rates due to differential activity of various DNA repair pathways[38–40]. In addition, we also included cohesin binding sites and CTCF motifs (which associate with transcription factor binding[41] and promoter-enhancer loops[42]), distance of DSB from 5′ gene end, gene length, CpG content, and finally copy number (CN), a known important determinant of non-specific Cas9 cut toxicity[9,16,43]. Finally, prompted by recent studies of balance between DNA-repair pathways repairing Cas9-induced DSB[35,44], we considered whether sgRNA target sites that promote microhomology-mediated end joining (MMEJ) repair have different toxicity from those who do not (whose main mechanism of generating indels would presumably be canonical NHEJ). In order to assess the relevance of the DSB repair pathway relative activity in p53-mediated break toxicity, we identified target sgRNAs with nearby DNA microhomology (MH), required for MMEJ (see Methods).

Because various active chromatin marks tend to co-occur with other epigenomic features such as early RT and high DHS, we employed a methodology based on negative binomial (NB) regression that can adjust for the confounding between variables, finding associations particular to each of the epigenetic features after conditioning on other considered features[45,46] (see Methods). More specifically, in order to identify covariates of TP53-mediated toxicity of sgRNAs we performed two complementary analyses: (i) interaction of TP53 status with each of the genomic features, to detect an increase of DSB toxicity in feature-rich regions when p53 is intact, and, as a supporting analysis, (ii) the toxicity of each feature per se in the TP53wt cells alone. In other words, this approach detects chromatin environments that increase or decrease DSB-related toxicity in presence of active p53 (see Methods).

The main results (Fig. 2a; see Supplementary Fig. 6a for the complete results) show that DSB in active open chromatin correlate with a higher p53 toxicity: considered individually, euchromatin features at sgRNA loci are significantly associated with increased fitness loss in presence of p53, consistently for the three independent A549 experiments. This includes the presence of the active chromatin features DNase, H3K27ac, H3K4me2, H2A.Z, H3K79me2, and H3K36me3; the latter two features are associated with transcription elongation at active gene bodies[47]. Consistently, higher mRNA expression levels also show this association, while presence of the heterochromatin markers H3K27me3, H3K9me3, and lamin B1 have the opposite effect. One possible explanation for this increase of p53-dependent DSB toxicity in euchromatin could be that Cas9 has lower cutting efficiency in heterochromatin[35]: in other words, sgRNAs targeting loci in active open chromatin might trigger higher overall toxicity due to higher DSB occurence rates and not because each DSB exerts more toxic effects. However, our data do not favor this explanation (see Supplementary Text 2b). Overall, the results support that p53 enhances Cas9 DSB toxicity in active chromatin in human cells, in agreement with a recent study[20]. Additionally, the higher CN of a target site is correlated with a larger p53-dependent toxicity, as in previous reports[16,27]. Remarkably, the effect sizes (correlation coefficients, averaged across pseudo-replicates) of several chromatin features considered herein, for instance the active transcription elongation marks H3K79me2 (−0.0272) and H3K36me3 (−0.0165), and DHS (−0.0376), are similar to the known toxic effect of high CN segments in our data (−0.0396). Using the known effects of CN gain as a unit of measurement for the toxic effects, we estimate that targeting H3K79me2, H3K36me3 and DHS regions would correspond to targeting a region with a 2-fold, 1.2-fold, or 2.8-fold local increase in ploidy, respectively (see Methods).

Finally, we considered the possibility that DSBs could exert additional toxic effects in a p53-independent manner. Indeed, there seem to be both a p53-independent and p53-dependent DSB toxicity of roughly similar magnitudes, with some differences in the local features associated, suggesting different underlying mechanisms. In both cases there are active chromatin features that are associated to a higher toxicity (Supplementary Fig. 7).

We also examined associations with all the considered variables combined in a pairwise manner, where each variable is conditioned upon each one of the others. The associations with TP53 status reported above remain overall significant except in some cases, e.g., the modest effect of DSB distance from 5′ gene end disappears when correcting for gene length (see Supplementary Text 2c). Of note, all regression analyses adjust for gene essentiality as a covariate (Demeter2 score[48], see Methods), thus the fitness effects observed are not attributable to loss of gene function, but are rather due to the DNA break itself and/or the repair thereof.

## DNA sequence motifs enrichment near high-p53-toxicity sgRNA target sites

Motivated by the known associations of DNA motifs, such as various types of repeats or CTCF binding sites, with DNA repair and mutagenesis[41,49,50], we hypothesized that certain DNA motifs may be commonly proximal to sites that elicit p53-related toxic effects. To investigate, we ran HOMER v3[51] to identify DNA motifs enriched in nearby target loci in comparison with background loci, considering three distance ranges centered at the Cas9 cut position (from 16 nt to 100 nt), while contrasting the TP53wt versus TP53$^{-/-}$ backgrounds (see Methods). Statistical testing on HOMER-prioritized motifs was performed by NB regressions of the sgRNA read counts, in the same fashion as above, based on the presence or absence of each candidate motif (HOMER $p < 1e-5$) from each HOMER analysis, testing for interaction with TP53 status. There were three motif clusters (see Methods) whose TP53 interaction was significant and consistent across the three A549 pseudo-replicate experiments (Fig. 2d, Supplementary Fig. 8a); a total of 21 motifs are included in these clusters by similarity criteria (HOMER score >0.6) (Fig. 2e, Supplementary Fig. 8b). For example, motif TCGCGGGGGA from motif cluster 1 encompasses the cut position in 9.69% of target but 5.91% of background loci (HOMER $p = 1e-10$; average interaction effect size −0.12); furthermore, the motifs in this cluster have an average starting position located at a distance of $8.13 \pm 1.34$ bp upstream from the cut site. In the same manner, motif clusters 2 and 3 each consist of similar motifs positioned at an average $25.52 \pm 9.98$ bp, and $50.77 \pm 26.73$ bp from the break, respectively. Of note, many of these motifs contain a CCCC segment prefixed or suffixed by one or more Gs (or a reverse complement thereof). Recent work reported a higher toxicity of high-GC content sgRNAs[52]. Our analysis solidifies this, proposing a TP53-dependent mechanism as well as nominating proximal DNA motifs consisting of consecutive Cs or Gs as correlates of Cas9 toxicity.

Similarly, we tested associations with the PAM sequence context of the target loci, contrasting against background loci (see Methods). We observed that a PAM sequence with a cytosine 1 bp upstream (5′-CNGG-3′) is strongly correlated with higher p53 toxicity (Fig. 2f; see also Supplementary Fig. 9a for all associations, and Supplementary Fig. 9b, c for the distributions of PAM trinucleotides and their contexts). The CGGG context was particularly toxic (average interaction effect size −0.22). Overall, we suggest that a careful Cas9 reagent design that avoids certain sequence patterns close to the sgRNA target locus could minimize p53 toxicity.

Additionally, we considered the p53-independent versus the p53-dependent fitness effects. The analyses show that a cytosine directly upstream of the PAM increases p53-independent DSB toxicity (x-axis; negative values indicate higher toxicity), and confirms that p53 increases this toxicity (y-axis) (Supplementary Fig. 10). However, a guanine upstream of the PAM also increases the p53-dependent toxicity (y-axis in Supplementary Fig. 10), while this guanine appears protective from p53-independent DSB toxicity.

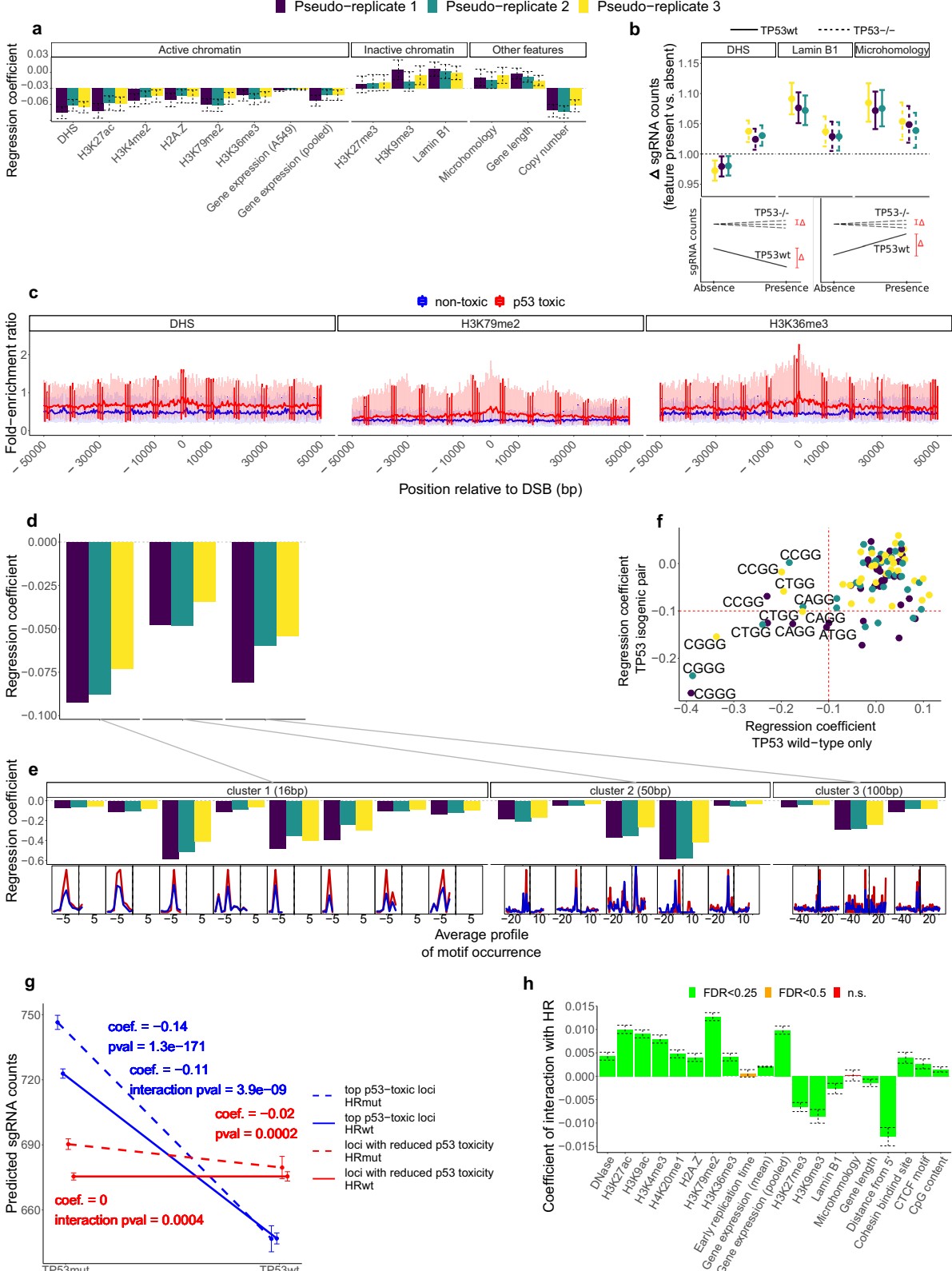

## Role of HR versus NHEJ repair pathway in Cas9 DSB toxicity across chromatin states

Recent studies have shown that Cas9 DSB, considered overall, are more toxic in TP53wt cells[13,15,16]. This was suggested to potentially cause problems in CRISPR genetic screenings, such as a reduced sensitivity for identifying gene essentiality[14,17,18]. Our data demonstrates that Cas9-induced p53 toxicity is not homogeneous genome-wide in a

manner associated with local chromatin features, DNA sequence motifs and microhomologies (see above). A recent study analyzed a panel of sgRNA target sites, and suggested that DSB repair pathway differential recruitment depends on chromatin accessibility[35]. We thus used genome-wide data sets to test the hypothesis that DSB-induced p53 toxicity at a genomic region depends on the DSB-repair mechanism. In particular, we made use of data from genetic screens

**Fig. 2 | Association of active chromatin marks with high-p53-toxicity sgRNA target sites. a** NB regression coefficients for each chromatin feature tested independently. Each colour represents one of the three pseudo-replicates. The regression coefficients are those from the interaction of *TP53* status with a given variable (see Supplementary Fig. 6a (bottom) for the effect of each variable per se, including only the TP53wt samples). Negative regression coefficients indicate a decrease of sgRNA counts. All regression coefficients have FDR < 0.25. $n = 10,050$ independent sgRNAs and 14 chromatin features were examined over six independent samples at three time points. Error bars represent the SE of the mean. **b** Interpretation of the regression coefficients of the interactions between *TP53* status and three selected features. There is a larger departure of the fitted sgRNA counts if the feature is present (its absence is scaled to 1) in TP53wt samples. For active chromatin feature DHS, the departure happens towards lower sgRNA counts in TP53wt samples (i.e., more p53 toxicity vinculated to presence of the feature), while the opposite is true for Lamin B1 (inactive chromatin) and microhomology. Error bars are 95% CI. Pseudo-replicates follow the same color scheme as in Fig. 1a. Schematics are included to aid interpretation of DHS (bottom left) and Lamin B1 and Microhomology (bottom right); the actual regressions are in Supplementary Fig. 6c. $n = 10,050$ independent sgRNAs and three chromatin features examined over six independent experiments. **c** Local abundance of a feature (represented as the ChipSeq fold-enrichment ratio), averaged at each 400bp-bin position relative to the sgRNA cut position (denoted 0), shown for the top 200 target loci exhibiting high p53 toxicity (larger negative LFC, red) and top 200 non-selected loci (LFC closer to 0, blue). Vertical lines represent the 25–75% interquartile range at each bin, and left-to-right lines connect the medians. Supplementary Fig. 6d shows the

corresponding figures when using another score. **d** Clusters of DNA sequence motifs identified by HOMER as enriched near target loci (FDR < 1e-5) −at different genomic distance to the sgRNA cut position− that show a significant (FDR < 0.25; red crosses indicate FDR > 0.25) and consistent association with higher p53 toxicity in the NB regressions (see Supplementary Fig. 8 for the effect of each variable alone regressed against the same sgRNA set, including only the TP53wt samples). **e** Separate regression results for motifs contained in the motif clusters. Below each motif are shown its relative frequencies at target (red) and background (blue) loci. The actual motif sequences are shown in Supplementary Fig. 8B. **f** Associations with all PAM sequence contexts, as regression coefficients. Top associations with DSB-related p53 toxicity are labelled. See Supplementary Fig. 9 for additional information. **g** Interaction of *TP53* and HR repair gene mutational status, using either the counts from the target loci (blue) or from the control loci (red). For the regressions including only HRmut cell lines (dashed lines) the regression coefficient and associated *p*-value are shown; for the regressions including only HRwt cell lines (full lines) the regression coefficient and *p*-value of the interaction of *TP53* and HR are shown. Error bars represent the 95% CI. $n = 16,174$ independent sgRNAs examined over 124 independent cell lines. **h** Regression coefficients of the interaction between each feature and the HR repair mutational status. Positive coefficients indicate that the increment of DSB toxicity when a feature is present (or more abundant) is alleviated in HRwt cells. Error bars represent the SE of the mean. FDR adjustment was performed to account for multiple comparisons. $n = 56,855$ independent sgRNAs and 20 chromatin features examined over 124 independent cell lines. Source data are provided as a Source Data file.

previously performed on 856 cell lines (Achilles 21Q2) to ask whether HR activity affects p53-dependent DSB toxicity, and if so, whether this occurs variably across open versus closed chromatin. We stratified the cell lines into HR deficient (HRmut) and proficient (HRwt) based on presence of deleterious variants in known HR genes, while restricting to four tissues-of-origin where HR deficiencies have been reported in tumors[53]. By analogy to the above analyses, we defined a set of 3,998 target loci (sgRNAs that potentially trigger more p53 toxicity; see Methods, and Supplementary Fig. 12) and a set of 8,625 sgRNAs with low p53 toxicity. We employed these two sgRNA sets to study the effect of the interaction between *TP53* and HR mutation status on DSB toxicity. Firstly, by using the target loci (Fig. 2g, blue), we observe an expected increase of DSB toxicity in TP53wt cell lines, which is however significantly less steep in HRwt than in HRmut cell lines (regression coefficient −0.11 versus −0.14, respectively; interaction term $p = 3.9e-9$); this suggests that DSBs repaired with HR are less p53-toxic. Secondly, by examining the set of loci with low p53-mediated toxicity (Fig. 2g, red), in the case of HRmut cell lines there is a slight p53 toxicity, however in the case of the HRwt cell lines, there is no p53 toxicity (interaction $p = 0.0004$). Considered together, these results suggest that the repair of Cas9 DSBs via the HR pathway triggers less toxicity than with other competing pathways; this may apply to the p53-associated toxicity as well as to other mechanisms of toxicity arising from Cas9 activity.

Next, we performed another analysis of the previous cell line panel data to interrogate whether this HR-dependent toxicity depends on the chromatin environment. In particular, we applied the NB regression method used in previous sections, to test statistical interactions between HR status and a chromatin feature, this time including library-wide sgRNAs (see Methods). This regression analysis supports the above observation that DSB toxicity is reduced in HRwt cell lines, moreso in active chromatin regions that we report as triggering more p53 toxicity, especially H3K79me2 (Fig. 2h).

In summary, our data is compatible with a mechanism where a DSB repair pathway other than HR repair contributes to Cas9 toxicity. This toxic mechanism is likely related with the canonical NHEJ pathway, because the alt-NHEJ promoted by microhomologies flanking the cut site appeared associated with protection from toxicity (see above). Supporting this hypothesis, the abundance of the key NHEJ protein XRCC4 peaks around the cut position of the p53-toxic compared to the

non-p53-toxic sgRNAs (Supplementary Fig. 16); this is also consistent with enriched chromatin marks around cut sites and their known links with various DSB repair mechanisms (discussed in Supplementary Text 2d). The canonical NHEJ-related mechanism either generates a higher amount of toxic intermediates when operative in active chromatin, or the amount generated is the same but their toxicity is higher in active chromatin. In addition to providing mechanistic insight, these statistical associations on a panel of 124 cell lines serve as a validation of the general link of p53 toxicity with active chromatin and other related features, as reported in our experimental dataset on the A549 isogenic pair.

## Discussion

Our data supports that Cas9 DSB-triggered toxicity mediated by p53 activity is a common occurrence in human cells[13,15,16]. p53-mediated toxicity is however highly variable depending on the locus targeted: both the DNA sequence and the epigenomic state in the region surrounding the target locus predict the fitness penalty of the cut. For instance, DSBs at sgRNA target sequences that are located in active, accessible chromatin trigger a stronger p53-toxic response, in agreement with a recent study[20]. These analyses can provide guidelines for choice of target sites for gene editing to minimize toxic effects; see Supplementary Text 2e for more on these guidelines and design of an example p53-toxicity score, which we applied to rank loci targeted in the human genome-wide screening libraries Brunello, TKO and Gecko (see Supplementary Dataset 2). Furthermore, our p53-toxicity score supports that there is a reduction of screening sensitivity when using sgRNAs with high p53-toxicity (Supplementary Text 2f). However, given that these genome-wide libraries were not specifically designed to measure the variation in toxicity of DSB in different chromatin environments, future experiments using a custom sgRNA library would allow a more comprehensive toxic/non-toxic sgRNA classifier to be developed.

Our analyses are in line with concerns that Cas9 activity in human cells, when used ex vivo or in vivo for therapeutic purposes, might select for *TP53*-mutant cells thereby having tumorigenic potential[13,20,22]. A judicious choice of loci targeted by gene editing reagents to minimize *TP53*-mediated toxicity would allay the concerns about such side effects. Another important application of our analyses involves the design of reagents for genetic screening libraries and

interpretation of data coming from screening experiments in TP53wt cells. We suggest that regions marked by certain active chromatin features such as DHS (associated with gene regulatory regions), H3K79me2 and H3K36me3 (associated with certain segments in transcribed gene bodies), would be advisable to avoid if possible, as well as the proximity of certain DNA motifs and the existence of off-target regions. Furthermore, while the mechanistic details underlying these associations remain to be elucidated, we suggest the exacerbated cut toxicity is at least partly due to preferential activity of different DSB repair pathways. Interestingly, among the chromatin marks that we found significant, H3K79 and H3K36 methylation were suggested to regulate HR repair and other repair pathways[36,37,54–56]. Our analyses suggest that use of MMEJ and HR pathways is associated with a less-pronounced TP53-mediated fitness loss; by elimination, we infer that canonical NHEJ or a related mechanism may be causal to the toxicity. In addition, HR is thought to be preferred for heterochromatic DSBs[57], which is compatible with our observation of a higher toxicity of DSBs in euchromatin. Use of Cas9 gene editing reagents might thus select for cells deficient in a particular DNA repair pathway such as NHEJ, resulting in compensation using other pathways such as MMEJ and HR. The latter can be mutagenic and recombinogenic, respectively, thus promoting genomic instability in Cas9-edited cells.

Next, it is conceivable that the DNA motifs we found associated with toxicity might reflect propensity to form various secondary structures and/or bind proteins that could interfere with normal processing of DSBs. Likewise, we found that a cytosine located upstream of NGG PAM sequences is associated with higher p53 toxicity, which might plausibly be due to differential Cas9 binding and/or activity dependent on the sequence near the PAM. A related issue are the DNA microhomologies near targeted sites: we propose that MMEJ may be a less toxic alternative compared to the competing DSB repair pathways, suggesting a potential utility of designing gene editing reagents to target regions enriched for microhomologies to promote MMEJ.

Incidentally, it has been shown that TP53 status may influence the expression of Cas9[13], potentially constituting a confounder in our analyses. However, an underexpression of Cas9 in TP53wt cells[13] should equally affect the efficiency of all sgRNAs in the library, i.e. it would not explain why only one or two of the sgRNAs targeting a gene has different effects between TP53 wild-type and mutant backgrounds. Likewise, lentiviral transduction efficiency could be hampered by p53 activity[58], however, since all sgRNA constructs are transduced via the same type of lentivirus, again all sgRNAs in the library should be equally affected. Finally, we also acknowledge that lentiviral integration has a preference towards active chromatin[59], however plausibly the integration site for a particular lentivirus DNA would not be correlated with the sgRNA sequence encoded within (and thus also the sgRNA target site) and so would not confound our analyses.

Finally, because the fitness effects due to DSB-related p53 toxicity are often variable between different sgRNAs targeting one gene in genome-wide screening libraries, p53 toxicity of DSB can thwart genetic screening experiments. We find TP53 wild-type cells to have an inflation of gene knockouts that decrease fitness, compared to their TP53−/− counterpart, although the assessment of gene essentiality is still possible[17,18]. However, we suggest these confounding effects become more acute when comparing gene essentiality between different conditions (e.g., control vs. drug treatment), which is a common type of experimental design of CRISPR screening assays. Overall, our analyses suggest additional principles that could be implemented to enhance the design of CRISPR screening libraries (and also to refine statistical methods for screening data analysis), in order to facilitate robust discovery of conditionally essential genes. Similarly, following the same design principles would reduce the toxicity of Cas9 activity to TP53 wild-type cells, minimizing concerns that gene editing might select for cells that predispose to genomic instability and cancer risk.

## Methods

### Cell culture and CRISPR-Cas9 screening

A549 cell line was generously provided by the Nebreda laboratory (IRB Barcelona), and it was authenticated by using an STR profile analysis. The cell culture, generation of doxycycline-induced cells, and generation of the TP53-isogenic pair of A549 cells was as described in our recent study, Biayna et al.[60], from where we quote verbatim: "A549 cell lines were [...] maintained with RPMI-1640 or DMEM medium and supplemented with 10% fetal bovine serum and 5% penicillin-streptomycin. [...] The NickaseNinja (ATUM, USA) vector co-expressing 2 gRNAs (pD1401-AD: CMV-Cas9N-2A-GFP, Cas9-ElecD) was used to generate the TP53 KO [...] cells. TP53 gRNA sequences (GCAGTCACAGCACATGACGG) (GATGGCCATGGCGCGGACGC) [...] were designed using the ATUM gRNA Design Tool. Moreover, 48 h post-transduction, positive GFP cells were sorted by FACS (FACSAria Fusion, BD, USA) and plated into 96-well plates. After 15 days, clones were collected and validated by western blot using the following primary antibodies: p53 (1:1,000; sc-47698, Santa Cruz Biotechnology); Vinculin (1:5,000; V9264, MilliporeSigma) [...]. Lentiviral particles were produced in HEK-293T cells using a pLKO.1-shRNA plasmid. The cell lines were transduced and selected with puromycin for 72 h [...]". Moreover the genetic screening data on the A549 TP53wt and TP53−/− cells (the pseudoreplicates with and without doxycycline) was as reported in Biayna et al.[60], from where we quote verbatim "For sgRNA screening of the A549 [...], A549^TP53−/− [...], cells were infected with the Brunello CRISPR Knockout Pooled Library (73179-LV, Addgene, USA)[31]. Infection with lentiviruses was performed at an MOI ≤ 0.4 for all cell lines. At 24 h postinfection, the medium was replaced with a selection medium containing puromycin (2 μg/mL). After 5 to 6 days of selection, cells were split into the different experimental conditions: [...] For A549 cell line, without and with DOX (3.9 μg/ml). All cell lines were passaged every 3 days (up to 15 days), and for each time point, the number of cells needed to maintain the predetermined coverage of 400- to 500-fold was taken. DNA extraction was performed using the DNA genomic Kit (Puregene Cell and Tissue Kit, Qiagen, Germany)"; note that in the current analysis we did not use previous data generated from APOBEC3A-overexpressing cells[60], and that we have performed additional experiments to screen both TP53 genetic backgrounds using a treatment combining doxycycline (DOX) and 0.5 μM of the ataxia telangiectasia and Rad3-related inhibitor (ATRi) AZD6738 (HY-19323, MedChemExpress) after the aforementioned five to six days of puromycin selection. A scheme of the experimental design is shown in Supplementary Fig. 11. NGS library preparation and sequencing was as described in Biayna et al.[60], from where we quote verbatim: "NGS libraries were prepared by 2-step PCR: For the first one, a total of 20 μg of DNA per a 12× reaction was used, and for the second PCR, a set of primers harboring Illumina TruSeq adapters as well as the barcodes for multiplexing were used [...]. Sequencing was carried out in the CNAG sequencing unit using 6 lanes of a 1 × 50 HiSeq".

### Data analysis

MAGeCK-VISPR[24] was used for alignment of the generated reads to the library, read counting, read count median normalization −based on (i) all sgRNAs, (ii) non-targeting control sgRNAs, or (iii) non-essential genes−, quality control (QC) analysis of the samples, and essentiality analyses using the (i) log₂ fold change (LFC) of sgRNA counts or (ii) maximum-likelihood estimation (MAGeCK-MLE) algorithms provided by MAGeCK-VISPR. In the case of MAGeCK-MLE, beta score significance p-value was estimated through 10 random sgRNA permutations, and a cutoff of false discovery rate (FDR) < 0.25 was applied. QC did not show a batch effect for the 12 and 15 days samples resulting from the sequencing arrangement. Each batch was treated independently in the normalization step, and the resulting normalized counts were either averaged or treated as technical replicates at each time point and treatment, depending on the analysis. Throughout the

figures, all boxplots are defined in the same manner, namely: the central line represents the median, the lower and upper box bounds represent the 25th and 75th percentiles, the upper and lower whiskers extend from their respective bounds no further than 1.5 times the inter-quartile range (IQR), and data points beyond the whiskers (i.e., outliers) are plotted individually.

## p53-mediated DSB toxicity does not prevent identifying universally essential genes

To evaluate how well the sgRNA dropout captures gene essentiality in the later time points, we used the R package pROC[61] to calculate the area under the receiver operating characteristic curve (AUROC) based on a list of 680 out of 684 essential[11] and 896 out of 927 non-essential[23] genes, sorted accordingly to the mean normalized sgRNA counts per gene. The same package was employed for comparing AUROC between *TP53* genotypes, using the bootstrap method with all defaults (1-tailed test). Mann-Whitney tests were also employed to compare pooled raw sgRNA counts between (i) essential and non-essential genes, and (ii) non-essential and 1000 control sgRNAs included in Brunello library that do not target any genomic region.

## *TP53* wild-type background can confound estimates of gene selection

MAGECK-MLE analyses consisted of 12 normalized sgRNA count level comparisons: each possible combination of a *TP53* genotype (wild-type or knockout), a condition (untreated, doxycycline-treated, or doxycycline and ATRi-treated), and a time point (9, 12, or 15 days of cell culture), compared to the corresponding time 0 sample. Beta score permutation FDR threshold for gene selection was 0.25. The untreated and doxycycline-treated samples could be considered pseudo-replicates 1 and 2, since no doxycycline-related toxicity nor conditional essentiality have been detected in A549[60]. The untreated and doxycycline-treated samples could be considered pseudo-replicates 1 and 2, since no doxycycline-related toxicity nor conditional essentiality have been detected in A549. Meanwhile, samples treated with both doxycycline and ATRi could be (conservatively) considered pseudo-replicates 3, bearing in mind that ATRi has been shown to be toxic by itself as well as in combination with gene inactivation[21]. The four sets of genes that, upon knockout, were systematically selected (either positively or negatively) in one of the *TP53* status and not in the other, across pseudo-replicates 1 and 2 and later time points (see Supplementary Table 3), were ascertained in the following way: for a gene to be considered systematically negatively selected only in TP53wt, this gene must be negatively selected in all pseudo-replicate TP53wt samples at later time points (t9, t12, and t15, where signal of selection would be more evident), while not being negatively selected in any corresponding TP53$^{-/-}$ sample. The top-50 *TP53* interactors were obtained from STRING[25] for all interaction sources and a minimum required interaction score of 0.9. We used the ANOVA function from the gdsctools[62] python package to compare pan-cancer essentiality data between wild-type and TP53mut cell lines from the combined projects Achilles and PScore, with an FDR < 0.25 threshold for conditional essentiality. We classified a cell line as TP53mut if it has at least one mutation that qualifies as pathogenic by passing both of these filters: (i) it has an allele frequency <0.001 according to gnomAD[63], and (ii) it is annotated as frameshift indel or nonsense, or as missense but it is present in at least three patients from three cohorts from the project GENIE[64] v10.1 (MSK-IMPACT341, MSK-IMPACT410, and MSK-IMPACT468), obtained from cBioPortal[65] (see Supplementary Dataset 1 for the classification of cell lines by mutation status). The nine other genes overlapping the ones identified in the A549 analyses were *NCL*, *CSNK1A1*, *HUWE1*, *NOB1*, *RHBDF1*, *RPF1*, *RPS29*, *HIRA*, and *SENP6*. We used GOrilla[30] to detect GO terms enriched with the set of negatively

selected genes, with the settings Two unranked lists of genes, and all gene ontologies (Process, Function, and Component).

## Overlap of confounded negative selection with public datasets

We used two public datasets consisting of *TP53*-isogenic pairs of the RPE1 cell line that employ different whole-genome CRISPR libraries: Brunello[14] and Gecko v2[19], respectively. Only the latest time point was included from each dataset (including our A549 dataset), as well as two replicates from each *TP53* status except in the case of the RPE1 Brunello-based dataset, in which one TP53wt replicate (R2 at day 28) was removed due to concerns about its quality. It is worth noting that this dataset has been considered to have quality issues such as poor editing efficiency[17,18]. The sgRNA count normalization used in these analyses was MAGeCK's median normalization based on the sets of non-targeting sgRNAs, because this inflates negative selection in TP53wt allowing for more negatively selected genes within each dataset replicate, and therefore more overlap between the different datasets. For the analyses we kept only the genes present both in Brunello and Gecko libraries (18,547 genes).

## *TP53* status can bias genetic screens for conditional essentiality

To detect gene essentiality conditional on ATRi, we compared the standardized beta score estimated by the MAGeCK-MLE algorithm (see *Data analysis* and *Differential gene selection patterns* above), calculating the mean beta score for the control (without doxycycline) and doxycycline-treated samples and contrasting them to the doxycycline +ATRi-treated samples. The top-20 ATRi-sensitizing genes were validated by at least one method, while the top-7 were validated by more than one method[21]: the latter are *POLE3*, *POLE4*, *KIAA1524/CIP2A*, *TYMS*, *C17orf53/HROB*, *TOPBP1*, and *APEX2*. The EMclust R package was employed for the expectation-maximization clustering step. We employed drugZ[33] and BAGEL v2[34] software to check whether the MAGeCK-MLE results are also replicated by other methods. drugZ was run with all defaults. The *normZ* score is a result of the integration of the different control and time points, and a negative *normZ* suggests conditional essentiality. BAGEL calculates fold change values from raw sgRNA counts, based on all time points for each treatment, and uses these to obtain the $\log_2$ bayes factor (BF) for each gene: namely, a positive BF indicates confidence that the gene is essential. For BF calculation, we used the core-essential and non-essential gene sets provided with the program as controls, and the default number of bootstrap iterations. BF values were averaged across control or doxycycline+ATRi samples, in the same manner as above. Feature abundance within *KIAA1524/CIP2A* was represented by the ChipSeq fold-enrichment ratio per 100 bp bin, in square root scale.

## Discovery of putative A549-specific ATRi-sensitizing genes is hampered in TP53wt

The randomization consisted on reshuffling 1000 times the hit/non-hit gene labels within each of the 12 possible comparisons that underlie the main analyses (see *TP53 status can bias genetic screens for conditional essentiality* above): the control was either a non-treated (without doxycycline) or doxycycline-treated sample (i.e., without beta score averaging), whose beta score were compared to doxycycline+ATRi-treated samples, for the six possible combinations of time point 9, 12, or 15 (again, without beta score averaging), and TP53wt or TP53$^{-/-}$. A gene was considered a hit when its normalized beta score had a value non-different from 0 in the control, lower than 0 in the ATRi-treated sample, and the difference of both beta score was also different from 0, with 2 SD as threshold of significance. We checked how often ≥2 hits occurred in ≥4 out of 6 TP53$^{-/-}$ and 0 out of 6 TP53wt comparisons, to approximate the probability that the two potential ATRi-sensitizing genes are not due to randomness: this occurred in 89 out of 1000 iterations.

## sgRNA-level analyses of cut toxicity reveal genomic and epigenomic determinants

For the following analyses, count normalization was based on the median of the set of non-essential genes recommended by the MAGeCK-VISPR authors[66], we merged the batches from each t12 and t15 sample previous to count normalization, and applied the remove-0 option. To restrict our analyses to a set of confidently negatively-selected sgRNAs, we focused on the LFC per individual sgRNA (instead of the average sgRNA LFC per gene) from the comparisons of counts between the A549 TP53wt (pseudo-treatment) and TP53$^{-/-}$ (pseudo-control) samples, for all treatments (control (without doxycycline), doxycycline-treated, and doxycycline+ATRi-treated) and later times (t9, t12, and t15); for example, TP53wt without doxycycline at t9 versus TP53$^{-/-}$ without doxycycline at t9. An sgRNA was considered to be negatively selected in TP53wt (relative to TP53$^{-/-}$) if LFC < −0.5 for all conditions (henceforth target loci). For the set of non-selected sgRNAs, we defined the threshold as |LFC| < 0.5. We removed target sgRNAs located within the top-50 TP53 interactors list from STRING[25], as well as within genes present in enriched GO terms. We also removed genes belonging to the families of olfactory receptor (OR), ubiquitin-specific peptidase (USP), and family with sequence similarity (FAM) genes, since gene families could be particularly enriched for paralogous sequences[67]. An extra analysis concerning the sgRNAs that are positively selected in TP53wt showed that there is not a consistent TP53 effect in all pseudo-replicates, suggesting that these positive selection signals could rather be experimental noise. This noise could be related to a lower Cas9 activity in TP53wt samples, as it has been recently suggested[13].

## Off-targeting enrichment at high-p53-toxicity sgRNA target sites

We employed Crisprseek[68] to calculate the total Cutting Frequency Determination[31] (CFD) scores for the sets of target and non-selected loci. Total CFD consists on the sum of the CFD scores of the top-5 off-targets. A target sgRNA was considered to have high off-targeting, and therefore removed, if it had any exact match, and/or it had a CFD above that from the 95 percentile non-selected locus (CFD = 3.14).

## Association of active chromatin features with high-p53-toxicity sgRNA target sites

The negative binomial (NB) regression analyses[45,46] have as dependent variable the sgRNA counts from all filtered (see previous steps) target loci, as well as from all the corresponding background loci (remaining sgRNAs in the same genes as the target loci), from sample pseudo-replicates 1 (untreated), 2 (doxycycline-treated), and 3 (doxycycline and ATRi-treated), at time points 9, 12, and 15. The explanatory variables were TP53 status (wild-type versus knockout) and several chromatin features and DNA motifs. These include, first, A549-specific data: (i) genome-wide chromatin mark ChipSeq abundance maps, obtained from RoadMap Epigenomics Consortium[69] (DHS, H3K9ac, H3K27ac, H3K4me1, H3K4me2, H3K4me3, H4K20me1, H2A.Z, H3K79me2, H3K36me3, H3K9me3, and H3K27me3), each classified into 0 (absence of feature) and 1 and 2 bins (increasing abundance of the feature, encompassing the same genome sizes), and with bin 1 removed to stress the differences between abundance and absence of a feature; (ii) Brunello library PAM sequences (with 1-bp context upstream or downstream); (iii) Presence versus absence of microhomology (MH): it is considered that two properties of the MH motifs greatly determine the rate of MMEJ recruitment: length of the motif, and position relative to the DSB, with 5−25 bp MH motif pairs both at 10 bp from the DSB position being optimal[70] −based on this, we applied a heuristic approach, consideringing an sgRNA as a candidate for MMEJ upon DSB if it contains at least one MH motif pair that is 5−15 bp long and within 15 bp from the DSB position; (iv) Long versus short genes (comprising two equally-sized bins after the removal of genes longer than 200 kb);

(v) Copy number data originally classified into 1 (low) to 7 (high) bins[71], further binarized into 5 to 7 versus 1 to 4; and vi) Other quantitative features: distance of DSB to 5′ gene end, scaled from 0 to 1; gene expression levels[72]; and number of CpG dimers in the target context sequence. Secondly, we also employed chromatin feature maps pooled across several cell lines[45], including (i) Replication time, originally classified into 0 (no replication) and 1 to 6 bins (later to earlier replication time), which were binarized into 6 (early) versus 1 to 5 (late). A few 0-bin sgRNAs were removed from the analysis, since we expect gene regions to replicate; (ii) Presence versus absence of motifs with CTCF and/or CTCF Cohesin; and (iii) High versus low gene expression levels. We also included presence versus absence of Lamin B1 (which indicates lamina associated domains) based on NKI Nuclear Lamina Associated Domains (LaminB1 DamID) tracks[73] from TIG-3 fibroblasts, obtained from genome.ucsc.edu. Finally, as a correcting variable we included the mean gene essentiality based on RNA inactivation across 712 cell lines from the DEMETER2 project[48] (D2). We used liftOver[74] to convert the genomic coordinates of Brunello sgRNA target sequences from hg38 into hg19, in order to match the build of the chromatin feature maps detailed above. See Supplementary Fig. 9b for the distribution of bins per feature. More generally, the NB regressions followed the model formula sgRNA raw counts ~ feature*TP53-status + D2 + offset, and were run using the glm.nb function from the MASS R package. The categorical variables had the default 'dummy' contrast when these were unordered, and orthogonal polynomial contrast (linear) when ordered. The offset was the natural logarithm of the total sum of non-targeting sgRNA counts per sample. In the case of the regressions using the isogenic pair (both TP53 status) we reported the estimates (regression coefficients) for the interaction between each feature (in general, presence vs. absence) and TP53 status (wild-type versus knockout), while in the other set of regressions we only included TP53wt samples (model formula: TP53wt sgRNA raw counts ~ feature + D2 + offset) and reported the estimates for each feature per se. In each of these cases, we first regressed each feature (interacting or not with TP53) separately, and then repeated the regression including each one of the remaining features (again, interacting or not with TP53) in a pairwise manner. We also ran two NB regressions including only either the TP53 status or D2 as explanatory variables, to capture the effect in toxicity of these features alone (model formulae: sgRNA raw counts ~ TP53-status + D2 + offset or sgRNA raw counts ~ D2 + offset). FDR values were calculated using the qvalue function from the qvalue R package. To represent the abundance of a feature surrounding the DSB position of the top 200 target loci (larger negative LFC) and top 200 non-selected loci (mean LFC closer to 0), we plot the ChipSeq fold-enrichment ratio (Fig. 2c) or -log$_{10}$ Poisson p-value (Supplementary Fig. 6d), averaged at each 400bp-bin position relative to the sgRNA cut position.

## Correspondence of p53-related DSB toxicity between copy-number and other chromatin features

One could use the known toxic effects of Cas9 DSBs at copy-number (CN) amplified loci to calibrate a unit of measurement for DSB toxicity due to various effects (e.g., chromatin states). The two levels of the CN (categorical) variable included in our regression analyses result from the binarization of the CNVkit score obtained for the A549 cell line exome (see the previous paragraph). In brief, this CN score represents the multiple of the variation from the overall ploidy, so that a CN score = 1 implies no local variation from the global ploidy. The mean CN score per level of the categorical variable was 0.7 (reference level) and 2.0 (high CN), respectively. Thus the regression coefficient beta = −0.0396 for CN could be interpreted broadly as a 2.9-fold increase in ploidy at a target locus resulting in an exp(−0.0396) = 0.96-fold decrease in sgRNA counts (note: 2.9 equals the ratio of 2.0 and 0.7, which are the CNs for the two levels of our categorical CN variable). Applying the simple rule-of-three (cross-multiplication), we infer that

targeting a genomic region marked by H3K79me2 incurs a toxicity that corresponds to targeting a locus with a ~2.0-fold increased ploidy. In the same manner, targeting a genomic region marked by either H3K36me3 or DHS would be analogous to targeting a region with an extra 1.2- or 2.8-fold ploidy, respectively.

## DNA sequence motifs enrichment near high-p53-toxicity sgRNA target sites

We ran three independent HOMER[75] analyses, one for each search length considered (16, 50, and 100 bp), centered on the DSB position. We used the hypergeometric test to find motifs between 2 and 25 bp long that were enriched in the set of target loci compared to the background loci (same sets as in previous analyses). NB regressions were performed in the same manner as in the previous point, including as explanatory variable the abundance versus absence of each motif with a HOMER $p$ <1e-5. We also included motifs that were similar or overlapped with the latter, using as threshold a HOMER similarity score >0.6, and that were likewise enriched in target loci (HOMER $p$ <1e-5); we referred to each group of similar motifs as a motif cluster. Regarding the abundance of a motif cluster in a locus, we only considered motif clusters represented once in at least 10% of all target and background loci, and only considered up to five instances of a given motif cluster in a locus. Motif clusters shown in Fig. 2d, e are those that pass both filters for significance and consistency: (1) the estimate of the interaction between the effect of a motif cluster and *TP53* has an FDR < 0.25 (Benjamini−Hochberg method), unless indicated by a red cross; and (2) the estimate sign is matched in the effect of the motif alone in the TP53wt-only analyses, for at least one pseudo-replicate. The plotMotifOccurrenceAverage function from the seqPattern R package was employed −with all defaults− to estimate the relative frequency of each motif (contained in the significant motif clusters) between the target and background loci; this analysis was based on the HOMER-inferred position weight matrices of the motifs. Motifs included in the motif cluster 1 that are not plotted in Fig. 2e (nor in Supplementary Fig. 8b) due to the difficulty or inability of *seqPattern* to find them, possibly because of their complexity, are BBCCCCG KDGWKKTK, CCDCCMCYGWTGWNDTY, CCMCHCHCGGGGGVC, KYC CYCGGHTKWYTB, NHCCHCMVNGGGGG, TCNCCACTGTKNDSWM, and YCHCCTCCGTGDDGT. The analysis that identified the most p53-toxic PAM sequence context is described in section *Association of active chromatin with high-p53-toxicity sgRNA target site* above.

## Role of HR versus NHEJ repair pathway in Cas9 DSB toxicity across chromatin states

We employed the raw sgRNA count data from the project Achilles[27] (21Q2). We computed the log₂ fold sgRNA counts averaged across the replicates of each cell line. We classified the 856 cell lines as either wild-type or mutated for *TP53* and HR pathways. We applied the algorithm detailed in section *TP53wt-conditionally essential genes* above, with the differences that for HR we considered that if any of the main genes of the pathway, this is, *BRCA1, BRCA2, PALB2*, and *RAD51C*, had a pathogenic mutation, or *BRCA1* contained a deletion[76], the HR repair pathway would be deficient. Also, a cell line was considered HRmut if it bears a rare missense variant with a CADD > 15 annotated with Annovar[77], and/or the cell line is classified as mutant for any of the four genes by a cancer functional event (CFE)[76]. In addition, only tissues considered to be typically BRCA-deficient were considered[78]; namely breast, ovary, pancreas, and prostate, leaving 124 cell lines for all the analyses. See Supplementary Dataset 1 for the mutation status assigned to each cell line. We employed gdsctools as detailed in section *TP53 wild-type background confounds estimates of gene selection* above. Specifically, we compared gene essentiality, measured as the relative sgRNA log₂ fold counts, between TP53wt and TP53mut cell lines. The filter applied to identify sgRNAs that potentially trigger more p53 toxicity (target loci) was to have a worse effect on fitness in TP53wt

cell lines, defined as a *gdstools* ANOVA effect size < −0.4; Supplementary Fig. 12 shows some examples of how this filter was applied. From the 5,382 sgRNAs that passed this filter, we removed 1384 based on criteria detailed in section *sgRNA-level analyses of cut toxicity reveal genomic and epigenomic determinants* above. A control set of low p53 toxicity loci was ascertained by setting a threshold of −0.04 < effect size < 0.04. To run the NB regressions, we employed 3,294 target and 6,316 low p53 toxicity loci. The model formulae applied were (i) sgRNA raw counts ~ *TP53*-status * HR-status + D2 + tissue + offset and (ii) sgRNA raw counts ~ feature * HR-status + D2 + tissue + offset. In (i), two independent regressions were run: one employing target loci, and another employing the loci with low p53 toxicity. In (ii), 56,855 sgRNAs (all the available ones from Avana library) were included. The chromatin feature maps employed are similar to those detailed above, and based on the pooling of several cell lines[45], including DHS, H3K9ac, H3K27ac, H3K4me3, H4K20me1, H2A.Z, H3K79me2, H3K36me3, H3K9me3, H3K27me3, CTCF motifs, replication time, gene expression levels, Lamin B1 association, CpG content, gene length, and sgRNA positioning with respect to 5′ versus 3′ gene ends. In both models, cell line tissue information was included as a correcting variable, with breast as reference level. The offset was based on the total read counts of 378 known non-essential genes[23].

### Reporting summary

Further information on research design is available in the Nature Research Reporting Summary linked to this article.

## Data availability

In this study published datasets were reanalyzed. We used the CRISPR screening data from a recent publication from our lab[60]. We also employed publicly available data as described in the Methods: In brief, we used pan-cancer essentiality data from Achilles and PScore projects[27–29], and gene essentiality information from the DEMETER2 project[48], available through DepMap repository [https://www.depmap.org]; CRISPR screening data of *TP53*-isogenic cell lines were obtained from previous publications[17,19] and from Taipale lab. A list of top-50 *TP53* interacting genes was obtained from STRING [https://string-db.org/]; Chromatin mark and DNase ChipSeq data was obtained from RoadMap Epigenomics Consortium [http://www.roadmapepigenomics.org/]; Copy number data was obtained from a recent publication from our lab[71]; Gene expression levels were obtained from a recent publication from our lab[72]; Replication time, CTCF and Cohesin motifs data was obtained from a previous publication[45]; Lamin B1 data was obtained from genome.ucsc.edu; Allele frequency data was obtained from gnomAD[63]; CFE (GDSC1) data was obtained from a previous publication[76]; Mutation data for MSK cohorts from the project GENIE66 v10.1 was obtained from cBioPortal [https://www.cbioportal.org/]; A list of validated ATRi-sensitizers was obtained from a previous publication[21]; Lists of core-essential[23] and non-essential[11] genes were obtained from previous publications. Raw sgRNA read count data generated in our lab and used in this study have been deposited in the Figshare database [https://doi.org/10.6084/m9.figshare.20326587.v1], while CRISPR data from other labs is available via the references cited. Source data are provided with this paper.

## Code availability

Custom code available in a github repository https://github.com/mmaalvarez/code_natcom_2022/tree/Alvarez_etal_2022_NatCom and Zenodo https://doi.org/10.5281/zenodo.6851052[79].

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

## Acknowledgements

Work in the F.S. lab was supported by the ERC Starting Grant "HYPER-INSIGHT" (grant number 757700 to F.S.); the Horizon2020 RIA project "DECIDER" (grant number 965193 to F.S.); the Spanish Ministry of Science, Education and Universities project "REPAIRSCAPE" (grant number PID2020-118795GB-I00 to F.S.); the State Agency for Research of the Ministry of Science and Innovation - Severo Ochoa Centre of Excellence Award (grant number CEX2019-000913-S to IRB Barcelona); and the CERCA Generalitat de Catalunya funds (to IRB Barcelona).

## Author contributions

M.M.A. designed the study, carried out all computational and statistical analyses, and wrote the manuscript. J.B. performed the CRISPR experimental work, and revised the manuscript. F.S. conceived, designed and supervised the study, and wrote the manuscript.

## Competing interests

The authors declare no competing interests.
