## [Peer Review File · Nature Communications]

Reviewers' Comments:

Reviewer #1:

Remarks to the Author:

Alvarez and others study the role of genomic loci and p53 status on CRISPR/Cas9 induced genotoxicity in human cells, mainly using isogenic pair of lung adenocarcinoma A549 cells.

They find that DSB induced toxicity varies based on chromatin environment of the site that is cut, limiting sensitivity of CRISPR/Cas9 screens. Furthermore, this effect is modulated by p53 status, highlighting the importance of using p53 null cells in high resolution CRISPR screens. Interestingly, it appears that the effect of p53 status depends on the repair pathway used, with NHEJ exhibiting the highest level of p53 dependent toxicity.

The work replicates many prior findings related to CRISPR/Cas9 induced genotoxicity and the role of p53 in increasing this toxicity. The novelty of the work relates to the finding that the level of the p53-dependent toxicity depends on features present on the genomic loci that is cut. The experiments are carefully done, and the findings reported are important. The manuscript is well written and clear. I recommend publications after minor revisions below.

major:

1) CRISPR/Cas9 induced DSBs are genotoxic, and p53 increases the toxicity. The authors should, however, make it clearer to the reader how much of the toxicity is p53 dependent and how much is not.

2) The detection limit of CRISPR screens is affected by the variance of guide effects. This problem is particularly acute in screens that target small genes or gene regulatory elements. It would be helpful if the authors quantitatively estimated the increase in sensitivity of one to five guide screens that can be achieved by p53 knockout and/or correction using their algorithm that takes into account the genomic locus cut.

3) In their guide-level analysis, the authors recover similar order of magnitude of guides that are differentially selected in p53 wt and p53 null cells. This is concerning, why are guides having stronger effect size in p53 wt cells? Are they targeting p53 pathway genes, or is most of the variation random noise?

minor:

Statement "One possible explanation for this increase of p53-dependent DSB toxicity in euchromatin could be that Cas9 has lower cutting efficiency in heterochromatin" is unclear. Why would this be?

Sentence "Remarkably, the effect sizes (averaged across pseudo-replicates) of several chromatin features considered herein, for instance the active transcription elongation marks H3K79me2 (-0.0272) and H3K36me3 (-0.0165), and the DNase hypersensitivity (-0.0376), are similar to the known toxic effect of high CN segments in our data (-0.0396): the relative percentage increase of CN is 46%, 140%, and 5%, respectively." is unclear, using simpler formulation, one could use copy number as the metric, stating that targeting open chromatin is equivalent to targeting a region with a copy number of x.

Reviewer #2:

Remarks to the Author:

In this manuscript, Alvarez et al. analyze CRISPR/Cas9 screens to explore TP53-dependent toxicity of CRISPR/Cas9 gene editing. The authors analyzed CRISPR/Cas9 screens performed with isogenic pairs of TP53-WT and TP53-/- A549 and RPE1 cells. Although TP53 status only modestly affected the ability to identify pan-essential genes, more genes showed significant negative and positive selection in TP53-WT vs. TP53-/- cells. The sgRNAs that resulted in stronger toxicity in TP53-WT

cells were enriched for features of active chromatin, i.e. increased fitness loss in the presence of p53 was associated with euchromatin features. In addition, p53-dependent toxic sgRNAs were enriched for specific DNA sequence motifs with high GC content. These results suggest that p53 enhances Cas9 DSB toxicity in active chromatin in human cells, and that avoiding certain sequences when designing sgRNAs might minimize p53-dependent toxicity. Moreover, some evidence is provided that, using current sgRNA libraries, the power to detect conditionally-essential genes is lower for TP53-WT cell lines compared to their TP53-/- counterparts (or, phrased differently, that TP53 status affects the ability to discover synthetic lethal gene interactions).

Overall, this is an interesting study that applies proper computational tools and statistical analyses to make incremental progress in our understanding of p53-dependent CRISPR/Cas9 toxicity. The degree of novelty is borderline for Nature Communications, in my opinion, and the manuscript reads quite technical in its current form (although it is clear for the most part). The Figures are also a bit technical and underwhelming as currently presented. On the other hand, the study does open intriguing directions for future research, and may have (few) specific practical implications. I would recommend the authors to try to rewrite the manuscript in a more succinct, biology-focused manner, as I believe that the study would be more compelling in a Nature-style Brief Communications format (perhaps with two Main Figures: one describing the effect of TP53 on essentiality screens, the other describing the context-dependence on open chromatin and DNA sequence).

A few specific comments and suggestions:

- (1) In addition to the sgRNA-dependent effect of TP53 status on DSBs and consequently on CRISPR screens, TP53 status may also influence the effects of Cas9 expression alone, as well as those of the lentiviral transduction used for genome-wide genetic screening. Can the authors analyze relevant data to address whether these potential TP53-related effects are also associated with the chromatin context, DNA sequence, etc.? This should also be mentioned in the 'Discussion' section.
- (2) Supplementary Fig. 1: If p53 activity exacerbates DSB toxicity, why is it that there are more counts of non-targeting sgRNAs in the TP53-WT A549 cells, but there is no difference in the count of the non-essential genes? Wouldn't the sgRNAs targeting non-essential genes expected to be more disadvantageous in the TP53-WT cells?
- (3) Lines 162-165: The number of sgRNAs that are negatively and positively selected in the TP53-WT cells is more or less equal (2,990 vs. 2,559, respectively). Shouldn't we expect more negative selection in the TP53-WT cells, if the key underlying mechanism is DSB toxicity?
- (4) Lines 351-364: Are the 57 genes that are differentially-essential in TP53-WT cells, and have no known direct association with TP53, have >1 sgRNA with TP53-conditional toxicity? This is the logical assumption, but it is not explicitly stated.
- (5) The section entitled "Association of early replicating and active chromatin with high-p53-toxicity sgRNA target sites" does not focus on replication timing (in fact, it barely mentions replication timing), so it seems more appropriate to rename it "Association of active chromatin with high-p53-toxicity sgRNA target sites".
- (6) All of the figures use pseudo-replicates of the A549 data, but I couldn't find a clear explanation of these pseudo-replicates – can the authors please clarify this in the main text?
- (7) Lines 325-326: "our data is compatible with a mechanism where a DSB repair mechanism other than HR repair contributes to Cas9 toxicity. This toxic mechanism is likely related with the canonical NHEJ pathway..." This is based on a comparison of HRwt vs. HRmut cell lines. Can a similar comparison be done between NHEJ-high vs. NHEJ-low cell lines (defined based on deleterious variants or transcriptional signatures)? This might provide a more direct evidence to the importance of NHEJ in this phenomenon (rather than just by way of elimination).
- (8) In most respects, the sections entitled "TP53 status biases genetic screens for conditional essentiality" and "Discovery of novel A549-specific ATRi-sensitizing genes is hampered in TP53wt" would better fit at the beginning of the manuscript, after describing the general effects of TP53 status on the identification of essential genes, and before diving into the chromatin- and sequence-dependencies. In other words, Fig. 5 could be re-located to become Fig. 2.
- (9) Throughout all of the Figures, the font is too small, making it difficult to read the axes-labels, etc.
- (10) Lines 460-462: "Both the DNA sequence and the epigenomic state in the region surrounding the target locus predict the fitness penalty of the cut, thus providing guidelines for choice of target

sites for gene editing to minimize toxic effects.” What are these guidelines exactly? In addition to the general discussion in this paragraph, it will be very helpful for the field if the manuscript provided (perhaps in a Supplementary Table or Supplementary Note) clear quantitative guidelines for the design of sgRNAs that are less affected by the TP53 status of the cells. Or perhaps a list of sgRNAs that are included in common libraries and should better be ignored when screening TP53-WT cell lines.

REVIEWER COMMENTS

Reviewer #1 (Remarks to the Author):

Alvarez and others study the role of genomic loci and p53 status on CRISPR/Cas9 induced genotoxicity in human cells, mainly using isogenic pair of lung adenocarcinoma A549 cells.

They find that DSB induced toxicity varies based on chromatin environment of the site that is cut, limiting sensitivity of CRISPR/Cas9 screens. Furthermore, this effect is modulated by p53 status, highlighting the importance of using p53 null cells in high resolution CRISPR screens. Interestingly, it appears that the effect of p53 status depends on the repair pathway used, with NHEJ exhibiting the highest level of p53 dependent toxicity.

The work replicates many prior findings related to CRISPR/Cas9 induced genotoxicity and the role of p53 in increasing this toxicity. The novelty of the work relates to the finding that the level of the p53-dependent toxicity depends on features present on the genomic loci that is cut. The experiments are carefully done, and the findings reported are important. The manuscript is well written and clear. I recommend publications after minor revisions below.

We thank the reviewer for their overall positive appraisal of our study, and also for their helpful suggestions and queries, which we answer below.

major:

1) CRISPR/Cas9 induced DSBs are genotoxic, and p53 increases the toxicity. The authors should, however, make it clearer to the reader how much of the toxicity is p53 dependent and how much is not.

We agree it would be useful to estimate the (relative) magnitude of the p53-dependent versus p53-independent toxicity. We have run a series of regressions analogous to the ones referred in “*Association of active chromatin with high-p53-toxicity sgRNA target sites*” section, the main difference being that this time all sgRNAs from the Brunello library were included, instead of a TP53-sensitive subset thereof (*target loci* in main text), allowing us to also capture other, p53-independent sources of DSB toxicity. To prevent gene essentiality from confounding the analyses, we included the Demeter score¹ in the regression as a covariate (as in main analyses: see Materials and methods).

The bottom barplot in the new Supplementary figure 7 indicates the chromatin feature associations with fitness effects (negative values denote a toxic effect, and vice versa) of a Cas9 double strand break (DSB) in TP53^{-/-} cells. In other words, this is an estimate of the DSB toxicity at a given chromatin feature, but without the participation of p53 (which was ko'd in this cell line). Meanwhile, the top barplot represents the coefficient of the “*feature:TP53*” interaction term, estimating how the DSB toxicity is altered in TP53 wild-type compared to TP53^{-/-} cells. This reflects the contribution of p53 activity to this Cas9 DSB toxicity (as analyzed in the original manuscript).

There seem to be both a p53-independent and -dependent DSB toxicity of roughly similar magnitudes, although interestingly with varying distributions across chromatin states. The p53-independent toxicity is positively associated with gene transcriptional activity: transcribed gene body marks H3K79me2 and H3K36me3 (toxic), high mRNA expression levels (toxic), and the facultative heterochromatin mark H3K27me3 (protective). The p53-dependent toxicity is associated with domain-scale features denoting euchromatin or heterochromatin: early replication time (p53-toxic), H3K9me3 (protective against p53-toxicity), and copy

number (p53-toxic, as was reported from prior work²⁻⁴). In addition, Lamin B1 proximity, and presence of microhomology near the site are protective from DSB toxicity both generally and with regards to p53-toxicity.

It is worth mentioning that we are more confident about the effect size estimates for the p53-dependent DSB toxicity component, than for the general (p53-independent toxicity component). The reason is the experimental design: our analyses are based on an isogenic cell line pair whose (presumably) only divergence is the TP53 status, hence the p53-dependent toxicity can be captured cleanly, rather than the overall DSB toxicity, which is potentially confounded by multiple agents.

Additionally, we considered the p53-independent versus the p53-dependent fitness effects in light of the DNA motif analysis (analogous to Figure 2D-E, where we considered the DNA motif association with p53-dependent toxicity using only the p53-dependent sgRNA subset). The enrichments of various PAM sequences (with and without immediate sequence context) show that a cytosine directly upstream of the PAM increases the p53-independent DSB toxicity (new Supplementary figure 10; x-axis; negative values indicate higher toxicity), and confirms the previously mentioned result that p53 increases this toxicity (new Supplementary figure 10; y-axis).

We have added these figures to the Supplementary material, and mentioned them in the main text.

2) The detection limit of CRISPR screens is affected by the variance of guide effects. This problem is particularly acute in screens that target small genes or gene regulatory elements. It would be helpful if the authors quantitatively estimated the increase in sensitivity of one to five guide screens that can be achieved by p53 knockout and/or correction using their algorithm that takes into account the genomic locus cut.

To address this topic, we made use of the A549 screening samples from our recent study Biayna et al. PLOS Biology 2021 (three pseudo-replicates, using mean gRNA counts between time points 9, 12, and 15). Namely, we calculated the AUC (for identifying known essential genes; see Materials and methods) using the full library (Brunello), and then also sequentially removing one, two, and up to three sgRNAs per gene, and checked the effect that this had on the AUC (i.e. the ability to differentiate core-essential from non-essential genes). The removed sgRNAs were either (a) the ones with the highest predicted p53 toxicity, or (b) the one with the lowest predicted p53-related DSB toxicity, according to a custom prioritization score that we defined in response to the 2nd Reviewer's 10th point.

Expectedly, the results show that, regardless of TP53 status, there is better accuracy if there are more sgRNAs targeting a gene. Importantly, the decrease in accuracy tends to be ameliorated if removing sgRNAs with the higher p53-toxicity according to our score (circle-shaped points in Supplementary Text 2F associated figure), and is exacerbated when the removed sgRNAs have the lowest p53-toxicity (triangle-shaped points). This effect is only found in TP53wt, also as hypothesized: in particular, in the case of TP53wt the mean AUC for the three pseudo-replicates after removing one, two, or three sgRNAs with the highest predicted p53-toxicity is 0.828, but if the removed sgRNAs are the ones with the lowest p53-toxicity the mean AUC is 0.814). As expected, In the case of TP53-/- the mean AUC in both scenarios is very similar, 0.818. Overall, this supports that the higher-toxicity sgRNAs have higher potential of confounding identification of essential genes in genetic screens, especially in a TP53wt environment.

We have added this analysis to the Supplementary Text 2F and referred to it in the Discussion.

3) In their guide-level analysis, the authors recover similar order of magnitude of guides that are differentially selected in p53 wt and p53 null cells. This is concerning, why are guides

having stronger effect size in p53 wt cells? Are they targeting p53 pathway genes, or is most of the variation random noise?

Indeed, we agree that it is unusual that there appear to be some genes/guides that crossed the threshold of (apparent) positive selection in the TP53 wild-type samples ('anti-toxic'), compared to the TP53^{-/-} samples.

We hypothesized that one possible explanation is that some of these positively selected guides may be false-positive calls (i.e. not really positively selected), based on the log-fold-change (LFC) threshold that we employed. To test this, we calculated for each sgRNA the ratio of the LFC mean and LFC variance between the three pseudo-replicate screens on the A549 cell line. Intuitively, true positives will show higher absolute effect sizes, and lower variance between pseudo-replicates, resulting in higher LFC mean-variance ratios, while false positives will have lower LFC mean-to-variance ratios.

Indeed, the negatively selected sgRNA set (*target loci* in the main text) has a significantly higher LFC mean-to-variance ratio than the corresponding positively selected set ($p=0.04$, Mann-Whitney test; see Supplementary figure 5). This suggests that part of the (apparently) positively selected set consists of false positives.

Furthermore, we asked if the reason for the existence of the positively selected sgRNA set could be that the functional consequences of gene function loss via k.o. (and not consequences of the DSB itself) are actually the reason for positive selection. In particular, inactivation of genes that participate in p53 function could contribute to increase cell fitness preferentially in TP53 wild-type cell lines. To investigate this, we checked whether a higher fraction of the positively selected sgRNAs that display the highest LFC mean-to-variance-ratios (>1.5 times the interquartile range, i.e. those guides with a higher signal-to-noise) have a p53-pathway gene as target, compared to the overall library ('Other') sgRNAs with highest LFC-ratios.

Indeed this was the case. The following tables show the percentages of sgRNA from each set that target top-50 p53-pathway genes (as defined via STRING database of functional interactions); and the subset thereof which are also known TSG according to TSGene database (bioinfo.uth.edu) and so are expected to have stronger effects.

	p53-pathway	No p53-pathway
Positively selected set	0.08%	9.94%
Overall library set	0.26%	89.72%

- OR = 2.78 (95% C.I.: 1e-3, 7885.02)

	p53-pathway that are also TSG		No p53-pathway
Positively selected set	0.02	4.42	
Overall library set	0.12	95.44	

- OR = 3.60 (95% C.I.: 1e-6, 11762344)

We have mentioned these possible explanations for the (apparently) positively selected gRNAs in the "sgRNA-level analyses of cut toxicity reveal genomic and epigenomic determinants" Section of the Results, and added the new Supplementary figure 5.

minor:

Statement "One possible explanation for this increase of p53-dependent DSB toxicity in euchromatin could be that Cas9 has lower cutting efficiency in heterochromatin" is unclear. Why would this be?

To clarify, we have added to the main text the following “: *in other terms, sgRNAs targeting loci in open chromatin may trigger higher overall toxicity simply due to the higher DSB occurrence rates and not because each of the DSBs exerts more toxic effects*”.

Sentence "Remarkably, the effect sizes (averaged across pseudo-replicates) of several chromatin features considered herein, for instance the active transcription elongation marks H3K79me2 (-0.0272) and H3K36me3 (-0.0165), and the DNase hypersensitivity (-0.0376), are similar to the known toxic effect of high CN segments in our data (-0.0396): the relative percentage increase of CN is 46%, 140%, and 5%, respectively." is unclear, using simpler formulation, one could use copy number as the metric, stating that targeting open chromatin is equivalent to targeting a region with a copy number of x.

We agree -- this is indeed a great point: one could use the known toxic effects of Cas9 DSBs at copy-number (CN) amplified loci to calibrate a “unit of measurement” for DSB toxicity due to various effects (e.g. chromatin states or DNA motifs as examined here).

The two levels of the CN (categorical) variable included in our regression analyses result from the binarization of the CNVkit score obtained for the A549 cell line exome (see Materials and methods). In brief, this CN score represents the multiple of the variation from the overall ploidy, so that a CN score = 1 implies no local variation from the global ploidy.

The mean CN score per level of the categorical variable “CN” was 0.7 (reference level) and 2.0 (high CN), respectively. Thus the regression coefficient $\beta = -0.0396$ for CN could be interpreted broadly as “a **2.9**-fold increase in ploidy at a target locus results in a $\exp(-0.0396) = \mathbf{0.96}$ -fold decrease in sgRNA counts” (note: 2.9 equals the ratio of 2.0 and 0.7, which are the CNs for the two levels of our categorical CN variable).

Applying the simple rule-of-three (cross-multiplication), we infer that targeting a genomic region marked by H3K79me2 incurs a toxicity that corresponds to targeting a locus with a **~2.0**-fold increased ploidy:

	Extra ploidy	Regression coefficient
CN	2.9-fold	-0.0396
H3K79me2	2 -fold	-0.0272

In the same manner, targeting a genomic region marked by either H3K36me3 or DHS would be analogous to targeting a region with an extra **1.2**- or **2.8**-fold ploidy, respectively.

Consequently, we have rephrased the last part of the text as: “*Using the known effects of CN gain as a “unit of measurement” for the toxic effects, we estimate that targeting H3K79me2, H3K36me3 and DHS regions would correspond to targeting a region with a 2-fold, 1.2-fold, or 2.8-fold increase in ploidy, respectively*”. We have also described the above calculations in the Methods.

Reviewer #2 (Remarks to the Author):

In this manuscript, Alvarez et al. analyze CRISPR/Cas9 screens to explore TP53-dependent toxicity of CRISPR/Cas9 gene editing. The authors analyzed CRISPR/Cas9 screens performed with isogenic pairs of TP53-WT and TP53-/- A549 and RPE1 cells. Although TP53 status only modestly affected the ability to identify pan-essential genes, more genes showed significant negative and positive selection in TP53-WT vs. TP53-/- cells. The sgRNAs that resulted in stronger toxicity in TP53-WT cells were enriched for features of active chromatin, i.e. increased fitness loss in the presence of p53 was associated with euchromatin features. In addition, p53-dependent toxic sgRNAs were enriched for specific DNA sequence motifs with high GC content. These results suggest that p53 enhances Cas9 DSB toxicity in active chromatin in human cells, and that avoiding certain sequences when designing sgRNAs might minimize p53-dependent toxicity. Moreover, some evidence is provided that, using current sgRNA libraries, the power to detect conditionally-essential genes is lower for TP53-WT cell lines compared to their TP53-/- counterparts (or, phrased differently, that TP53 status affects the ability to discover synthetic lethal gene interactions).

Overall, this is an interesting study that applies proper computational tools and statistical analyses to make incremental progress in our understanding of p53-dependent CRISPR/Cas9 toxicity. The degree of novelty is borderline for Nature Communications, in my opinion, and the manuscript reads quite technical in its current form (although it is clear for the most part). The Figures are also a bit technical and underwhelming as currently presented. On the other hand, the study does open intriguing directions for future research, and may have (few) specific practical implications. I would recommend the authors to try to rewrite the manuscript in a more succinct, biology-focused manner, as I believe that the study would be more compelling in a Nature-style Brief Communications format (perhaps with two Main Figures: one describing the effect of TP53 on essentiality screens, the other describing the context-dependence on open chromatin and DNA sequence).

We thank the reviewer for appreciating the interest of our study (e.g. with regard to opening directions for future research, and to practical implications), as well as for highlighting the proper application of computational tools and statistical analyses. We agree that the text and the figures had some level of technical detail in the original manuscript, and that they could be made more succinct. To this end, we have (a) relegated multiple sections of the text to either Material and Methods, Supplementary Figure legends, or a Supplementary Text, leaving a short summary of each section in the main text, and (b) reorganized main figures so that they follow the 2-figure organization as suggested by the Reviewer: the new Figure 1 describes the effect of TP53 genetic background on fitness screens, while the new Figure 2 shows the context-dependence of DSB toxicity on open chromatin and DNA sequence features. Please find below the list of sections of the text that have been moved to the new Supplementary Text or Methods:

- Old Fig. 3A-B bottom barplots (the p53wt-only analyses), moved to new Supplementary figure 8
- Old Fig. 4 panel A, moved to new Supp. fig. 12
- Old Fig. 5 panels B (moved to new Sup fig. 4C), and C-D (moved to new Sup. Text 1B)
- Results subsection "Overlap of confounded negative selection with public datasets" moved to Sup. Text 1A
- Results subsection "Discovery of novel A549-specific ATRi-sensitizing genes is hampered in TP53wt" moved to Sup. Text 1C
- Results subsection "Off-targeting enrichment at high-p53-toxicity sgRNA target sites" moved to Sup. Text 2A

A few specific comments and suggestions:

(1) In addition to the sgRNA-dependent effect of TP53 status on DSBs and consequently on CRISPR screens, TP53 status may also influence the effects of Cas9 expression alone, as well as those of the lentiviral transduction used for genome-wide genetic screening. Can the authors analyze relevant data to address whether these potential TP53-related effects are also associated with the chromatin context, DNA sequence, etc.? This should also be mentioned in the 'Discussion' section.

Indeed, the TP53 status may influence the expression of Cas9⁵. However, this phenomenon should equally affect the efficiency of all the sgRNAs, i.e. it would not explain why only one (or, at most, two) of the sgRNAs targeting a gene is differentially toxic between TP53 wild-type and -/- status, and the remaining sgRNAs are not. In contrast, the chromatin environment does vary across guides targeting the same gene, and could therefore explain these differences.

Lentiviral transduction efficiency may also be potentially influenced by the TP53 status⁶. However, we consider this issue to be addressed by the removal of the non-transduced cells via puromycin selection in the experiment (see Materials and methods), and moreover the non-transduced cells (even if they remained in the pool) would not contain sgRNAs and will not contribute any PCR product to the sequencing data pool.

We have briefly addressed this issue in the Discussion.

(2) Supplementary Fig. 1: If p53 activity exacerbates DSB toxicity, why is it that there are more counts of non-targeting sgRNAs in the TP53-WT A549 cells, but there is no difference in the count of the non-essential genes? Wouldn't the sgRNAs targeting non-essential genes expected to be more disadvantageous in the TP53-WT cells?

Firstly, considering that DSBs trigger p53 activity⁵, it is reasonable to predict that the absence of DSB will be particularly advantageous (in relative terms, compared to occurrence of DSBs) in a p53-expressing genetic background. Secondly, non-targeting sgRNAs do not cause any DSB in the DNA. Therefore, the observation that non-targeting sgRNAs have higher normalized read counts in wild-type cells, compared to TP53-/- cells, would be expected. In other words, cells expressing a non-targeting sgRNA will constitute a larger proportion of the wild-type cell pool than of the TP53-/- cell pool.

In contrast, sgRNAs that target non-essential genes do still cause DSB. However, the toxicity of a DSB is not different be it located in a non-essential gene or in any other gene. Therefore, **within each of the two TP53 backgrounds** the sole advantage of sgRNAs that target non-essential genes is the lack of gene function loss, so these sgRNAs should be equally selected without regard to p53 expression. In other words, we should not expect that cells with DSB in non-essential genes constitute a larger proportion of the wild-type cell pool than of the TP53-/- cell pool.

In summary, we think that these considerations can help to explain the observed lack of difference in non-essential gene sgRNA counts between wild-type and TP53-/- cells.

(3) Lines 162-165: The number of sgRNAs that are negatively and positively selected in the TP53-WT cells is more or less equal (2,990 vs. 2,559, respectively). Shouldn't we expect more negative selection in the TP53-WT cells, if the key underlying mechanism is DSB toxicity?

This was addressed in the response to the 1st reviewer's 3rd point; see above for detailed response. In brief, a part of this (apparent) positive selection seems to be spurious, attributable to noise: the positively selected sgRNAs have an overall lower signal-to-noise ratio ($p = 0.04$). In addition, we also noted in this set an enrichment of sgRNAs targeting p53-pathway genes (O.R. = 2.78), thus there may be positive selection on the downstream effects of abolishing gene function (rather than on the Cas9 DSBs themselves) causing a part of this apparent positive selection signal.

(4) Lines 351-364: Are the 57 genes that are differentially-essential in TP53-WT cells, and have no known direct association with TP53, have >1 sgRNA with TP53-conditional toxicity? This is the logical assumption, but it is not explicitly stated.

Among the 61 genes (i.e. considering *MDM2*, *MDM4*, *USP7*, and *AURKA* together with the 57 genes mentioned in the question) whose k.o. is differentially essential between TP53-WT and TP53-/- cells (namely, more negatively selected in TP53-WT cells), 28 genes either (i) are in a p53-associated pathway (according to the STRING database), and/or (ii) overlap with the hits from the Project Achilles⁷ dataset (which uses a different CRISPR library, Avana), and/or (iii) belong to an enriched biological pathway from Gene Ontology (GO), meaning that function loss is likely to be the cause of the differential selection of these 28 genes.

However, the cause of the differential selection of the remaining 33 genes may be related to other features, such as a differential toxicity of the DSB positions, which is determined by the sgRNA's target sequence. As the Reviewer suggests, a high presence of the sgRNAs that have the strongest negative selection in TP53-WT compared to TP53-/- cells ("p53-toxic sgRNA") would be expected in these 33 genes, given that they are negatively selected in TP53-WT cells. Seven out of these 33 genes actually have a sgRNA labeled as a "p53-toxic sgRNA", while the others do not have such a gRNA.

One possible explanation for this is that two different algorithms were employed: MAGECK-MLE for defining the 61 genes set, and the log₂ fold change (LFC) cutoff for the guide-level analysis. A key difference is that the former downweights outlier sgRNAs, so the effect of p53-toxicity in one sgRNA could be partially adjusted by the MAGECK algorithm (just as other sources of noise would be).

In addition, a sgRNA's mean LFC represents how much the cell fitness increases (positive values) or decreases (negative values) when a cell that is transduced with this sgRNA is TP53-WT instead of TP53-/-: therefore, the mean LFC values were used as a filtering score for the selection of p53-toxic sgRNAs (sgRNAs that have the strongest negative selection in TP53-WT compared to TP53-/- cells). Interestingly, the mean LFC values are indeed lower among the above-mentioned 33 genes (i.e. those whose loss of function is not likely to be the cause of their differential selection in TP53-WT cells) than in the remaining library, as shown in Response to reviewer Figure R1 below.

This observation supports the main point we would like to make here: that the gene-level (MAGECK-MLE) and guide-level (LFC threshold) analyses are not contradictory, but that the mean LFC threshold employed to select the p53-toxic sgRNAs was very strict. In other words, the 33 genes (out of the 61 genes that are more negatively selected in TP53-WT cells) that cannot be explained by gene function loss, are indeed targeted by sgRNAs with higher p53-related DSB toxicity than the remaining library (Fig. R3 below), but this is not immediately evident if we focus only on the top set of "p53-toxic sgRNAs", as the cutoff to define this set was very stringent.

We have discussed these considerations in the "sgRNA-level analyses of cut toxicity reveal genomic and epigenomic determinants" Subsection of the Results.

Response to reviewer Figure R1. Comparison of the mean sgRNA LFC values (y-axis) between gene sets. 28/61 false positives: their k.o. is more negatively selected in TP53 wild-type cells likely because of gene function loss; 33/61 potentially positive: more negatively selected in TP53 wild-type potentially due to the chromatin environment of the sgRNA target sequences; and the remaining genes in the library: 19053 out of 19114.

(5) The section entitled “Association of early replicating and active chromatin with high-p53-toxicity sgRNA target sites” does not focus on replication timing (in fact, it barely mentions replication timing), so it seems more appropriate to rename it “Association of active chromatin with high-p53-toxicity sgRNA target sites”.

We thank the reviewer for pointing this out. We have renamed the section “Association of active chromatin with high-p53-toxicity sgRNA target sites”.

(6) All of the figures use pseudo-replicates of the A549 data, but I couldn’t find a clear explanation of these pseudo-replicates – can the authors please clarify this in the main text?

In the Materials and methods section, the subsection “TP53 wild-type background inflates estimates of gene selection” explains what are the pseudo-replicates, and we here quote: “The untreated and doxycycline-treated samples could be considered pseudo-replicates 1 and 2, since no doxycycline-related toxicity nor conditional essentiality have been detected in A549. Meanwhile, samples treated with both doxycycline and ATRi could be (conservatively) considered pseudo-replicates 3, bearing in mind that ATRi has been shown to be toxic by itself as well as in combination with gene inactivation.”

As per the reviewer’s recommendation, we have added a brief explanation also to the main text (first paragraph in Results): “We used three biological pseudo-replicates, which differ in their treatment: either untreated, doxycycline-treated, or ATRi-treated (see Materials and methods for further details)”

(7) Lines 325-326: “our data is compatible with a mechanism where a DSB repair mechanism other than HR repair contributes to Cas9 toxicity. This toxic mechanism is likely related with the canonical NHEJ pathway...” This is based on a comparison of HRwt vs. HRmut cell lines. Can a similar comparison be done between NHEJ-high vs. NHEJ-low cell lines (defined based on deleterious variants or transcriptional signatures)? This might provide a more direct

evidence to the importance of NHEJ in this phenomenon (rather than just by way of elimination).

This is indeed an interesting question. However, an analysis analogous to that employed in the case of HR repair (drawing on HR-deficient cancer cell lines) is not feasible, since NHEJ deficiencies are not recognized to commonly occur in cancers nor cell lines, unlike HR deficiencies. Thus it is not clear whether the occurrence of mutations in NHEJ genes (e.g. *XRCC4*) is an effective proxy of NHEJ inactivation. Therefore, we followed a different approach to shed some light on the matter.

Firstly, there is recent evidence that H3K4 methylation locally promotes NHEJ by blocking end resection⁸. Our data (the top barplot in Supplementary figure 6A) shows that there is increased p53-related DSB toxicity in H3K4-methylated regions, consistent with a higher p53-related DSB toxicity of NHEJ repair. Furthermore, the same study claims that H3K4me3 as opposed to H3K4me1 is the main promoter of RIF1 accumulation (which contributes to blocking end resection); our above-mentioned analysis (top barplot in Supplementary figure 6A) shows that p53-related DSB toxicity increases with the methylation levels (H3K4me1 lowest, H3K4me2 and H3K4me3 highest), consistent with a higher p53-related DSB toxicity of RIF1 accumulation as a proxy of NHEJ repair. Overall, this association with histone marks supports that NHEJ repair triggers more p53-related DSB toxicity than mechanisms that rely on DNA resection, such as HR and MMEJ.

Secondly, it has been stated that, generally, “damage in active chromatin undergoes preferential repair via HR”⁹. Taking this into consideration, our result showing that active chromatin is less toxic by itself, i.e. independently of TP53 activity (Supplementary figure 7, bottom panel; see Response to reviewer 1 point 1) would be in accordance with a higher toxicity of NHEJ.

Finally, Chip-Seq normalized read count data for the key NHEJ protein *XRCC4*⁹ shows a peak of this protein around the cut position of the p53-toxic compared to the non-p53-toxic sgRNAs; see the Supplementary Text 2D associated figure. This plot is analogous to Figure 2C (in brief: normalized read counts were averaged at each 400bp-bin position relative to the sgRNA cut position (at 0), including the top 200 target loci showing more p53-related DSB toxicity – larger negative LFC, red – and top 200 non-selected loci – LFC closer to 0, blue. Vertical lines represent the 25-75% interquartile range at each bin, and left-to-right lines connect the medians). This *XRCC4* enrichment at the toxic DSB sites compared to non-toxic sites implicates NHEJ in toxicity.

We think that, considered together, these points further support the hypothesis that NHEJ repair of DSB results in higher p53-related toxicity compared with other competing DSB repair mechanisms.

We have included a discussion of this matter in Supplementary Text 2D and referred to it in the Discussion.

(8) In most respects, the sections entitled “TP53 status biases genetic screens for conditional essentiality” and “Discovery of novel A549-specific ATRi-sensitizing genes is hampered in TP53wt” would better fit at the beginning of the manuscript, after describing the general effects of TP53 status on the identification of essential genes, and before diving into the chromatin- and sequence-dependencies. In other words, Fig. 5 could be re-located to become Fig. 2.

We agree and have addressed this in the new arrangement of the Figures and text: the old Figure 5A is now the new Figure 1E (the remaining panels from the old Figure 5 are now in the Supplementary material), and the main text has been also rearranged accordingly. We thank the reviewer for the suggestion.

(9) Throughout all of the Figures, the font is too small, making it difficult to read the axis-labels, etc.

We have addressed this issue in the revised Figures.

(10) Lines 460-462: “Both the DNA sequence and the epigenomic state in the region surrounding the target locus predict the fitness penalty of the cut, thus providing guidelines for choice of target sites for gene editing to minimize toxic effects.” What are these guidelines exactly? In addition to the general discussion in this paragraph, it will be very helpful for the field if the manuscript provided (perhaps in a Supplementary Table or Supplementary Note) clear quantitative guidelines for the design of sgRNAs that are less affected by the TP53 status of the cells. Or perhaps a list of sgRNAs that are included in common libraries and should better be ignored when screening TP53-WT cell lines.

We agree that the criteria we found to influence p53-related toxicity of DSB could be summarized into a convenient quantitative score. To do so, we have assigned a "p53-toxicity score" to every sgRNA in the widely-used Brunello, Avana, TKO v1 and v2, and Gecko libraries (new Supplementary Table 7). This score consists of a combination of several of the most relevant variables associated with such toxicity: presence of Lamin B1 and microhomology (MH), binarized (as described in Materials and methods) GC content, cutting frequency determination (CFD), abundance of DHS, H3K9me3, copy number (CN), and replication time (RT).

The contribution of each feature to the p53-toxic phenotype score is the exponentiated regression coefficient of its interaction with TP53 status, averaged across the three A549 pseudo-replicates. This bears some resemblance with the analysis done to address Reviewer 1 point 1 (see above), with some distinctions: i) no sgRNA filtering based on the Cutting Frequency Determination (CFD; measure of the off-targeting effect of a sgRNA target sequence) score was applied, and ii) all the interaction terms were run in the same regression, thus ensuring that each association is conditioned upon the other factors (which may be correlated), i.e.

$$\text{sgRNA counts} \sim \text{DHS} * \text{TP53status} + \text{H3K9me3} * \text{TP53status} + \text{RT} * \text{TP53status} + \text{LaminB1} * \text{TP53status} + \text{CN} * \text{TP53status} + \text{MH} * \text{TP53status} + \text{GCcontent} * \text{TP53status} + \text{CFD} * \text{TP53status} + \text{D2score} + \text{offset}$$

Please note that we also included in the regression above the features i) presence of a C upstream of the PAM, and ii) H3K79me2 abundance. However, their interaction coefficients were not <0 (i.e. no toxic effect in this particular analysis, possibly due to correlations with other features) and were thus not included in the final score.

feature	contribution (exp(regression coefficient))
CN	0.941
DHS	0.970
GC content	0.979
RT	0.979
CFD	0.998
H3K9me3	1.01
Lamin B1	1.01
MH	1.01

A contribution < 1 implies a toxic effect (lower sgRNA counts) for the presence/abundance of a given feature, and *vice versa*. The p53-toxicity score for each sgRNA was calculated as follows:

$$\text{p53-related DSB-toxicity score} = \prod_{\text{features}} \text{bin} \times (\text{contribution} - 1) + 1$$

where a feature's bin is either 0 or 1. The scores were then rescaled so that the most p53-toxic sgRNA across libraries is assigned a 1, and the least p53-toxic one is assigned a 0.

To gauge the accuracy of this empirical "p53 toxicity score", we assigned the corresponding score to each sgRNA in Brunello library, and compared the distributions between the sets of sgRNAs that are strongly

negatively (“target loci” in text) or positively selected in TP53 wild-type **compared to TP53^{-/-} cells**, or not strongly selected to either side.

Response to Reviewer Figure R2. Comparison of p53-related DSB toxicity between sgRNA sets.

Mann-Whitney tests (one-tailed) show that the known negatively selected set of sgRNA has higher p53 toxicity phenoscores than normal sgRNAs ($p = 1.7e-13$), and the positively selected set has the lowest scores ($p < 2.22e-16$). This supports that our p53 toxicity score seems to correctly capture the p53-dependent toxicity triggered by DSB.

Going forward, we think a custom-designed library would be better suited to investigate effects of chromatin on DSB toxicity. Since the Brunello library is not specifically designed to measure the variation in toxicity of DSB in different chromatin environments, we think experiments using a custom sgRNA library would allow a more comprehensive toxic/non-toxic sgRNA classifier to be developed.

We have described this analysis in the Supplementary Text 2E.

LITERATURE CITED

1. McFarland, J. M. *et al.* Improved estimation of cancer dependencies from large-scale RNAi screens using model-based normalization and data integration. *Nat Commun* **9**, 4610 (2018).
2. Tzelepis, K. *et al.* A CRISPR Dropout Screen Identifies Genetic Vulnerabilities and Therapeutic Targets in Acute Myeloid Leukemia. *Cell Reports* **17**, 1193–1205 (2016).
3. Aguirre, A. J. *et al.* Genomic Copy Number Dictates a Gene-Independent Cell Response to CRISPR/Cas9 Targeting. *Cancer Discov* **6**, 914–929 (2016).
4. Munoz, D. M. *et al.* CRISPR Screens Provide a Comprehensive Assessment of Cancer Vulnerabilities but Generate False-Positive Hits for Highly Amplified Genomic Regions. *Cancer Discov* **6**, 900–913 (2016).
5. Enache, O. M. *et al.* Cas9 activates the p53 pathway and selects for p53-inactivating mutations. *Nat Genet* (2020) doi:10.1038/s41588-020-0623-4.
6. Piras, F. *et al.* Lentiviral vectors escape innate sensing but trigger p53 in human hematopoietic stem and progenitor cells. *EMBO Mol Med* **9**, 1198–1211 (2017).
7. Meyers, R. M. *et al.* Computational correction of copy number effect improves specificity of CRISPR–Cas9 essentiality screens in cancer cells. *Nat Genet* **49**, 1779–1784 (2017).
8. Bayley, R. *et al.* H3K4 methylation by SETD1A/BOD1L facilitates RIF1-dependent NHEJ. *Molecular Cell* S1097276522002684 (2022) doi:10.1016/j.molcel.2022.03.030.
9. Clouaire, T. *et al.* Comprehensive Mapping of Histone Modifications at DNA Double-Strand Breaks Deciphers Repair Pathway Chromatin Signatures. *Molecular Cell* **72**, 250-262.e6 (2018).

Reviewers' Comments:

Reviewer #1:

Remarks to the Author:

The authors have addressed my concerns well, I recommend publication of the work. I have only one suggestion for clarification: the authors only address the positively selected guides in response to my last point and the other reviewers point 3. Could they clarify also the case regarding negatively selected guides? Could there be more of them because p53 increases reproducible variance due to the chromatin locus targeted, which increases the number of hits? Or is there another explanation?

Reviewer #2:

Remarks to the Author:

The authors have addressed most of my concerns. The structure and of the paper is much better now, and it is easier to read. I appreciate that the authors have made a considerable effort to address the points that were raised, and in most cases they have done so successfully. Of particular value is the new Supplementary Table 7, which reports the p53-related DSB-toxicity scores for several widely-used CRISPR screening libraries. I therefore support the publication of this paper in Nat Commun.

I have a couple of remaining minor comments:

- 1) Response to point #1, the authors dismiss the potential effect of lentiviral transduction and Cas9 expression on the p53-dependent sgRNA toxicity. I am not convinced that these factors are irrelevant – both the effect of lentiviruses and that of Cas9 expression are likely to preferentially affect specific genomic loci, which may differentially affect different sgRNAs. I think the Discussion paragraph that deals with this topic should acknowledge this.
- 2) Response to point #7: The Chip-Seq analysis of XRCC4 is very nice. The association with genomic features, however, is indeed consistent with the hypothesis but is very circumstantial and does not provide any direct support a role for NHEJ in the described phenomenon.
- 3) The paper by Sinha et al. (Nat Commun 2021) should be mentioned/cited in the context of the comparison of CRISPR screens between TP53+/+ and TP53-/- cells.

REVIEWERS' COMMENTS

Reviewer #1 (Remarks to the Author):

The authors have addressed my concerns well, I recommend publication of the work. I have only one suggestion for clarification: the authors only address the positively selected guides in response to my last point and the other reviewers point 3. Could they clarify also the case regarding negatively selected guides? Could there be more of them because p53 increases reproducible variance due to the chromatin locus targeted, which increases the number of hits? Or is there another explanation?

The number of sgRNAs that are especially negatively selected in TP53wt cells (2,990) is determined based on a chosen log fold-change (LFC) cutoff for the TP53wt vs. TP53-/- comparison: a higher absolute value would result in smaller subsets.

Regarding these 2,990 sgRNAs that are especially negatively selected in TP53wt cells, the proposed mechanism of selection (main finding of our study) is that they target genomic regions with presence of e.g. active chromatin marks, and/or absence of inactive chromatin marks (see Results).

We do not think that this number is high because *TP53* status affects reproducible variance in the data. In particular, panel B in Supplementary figure 5 shows the LFC variance across pseudo-replicates: one-tailed Mann-Whitney tests indicate that variance in negatively-selected sgRNAs is in fact lower than in the remainder of the library ("Other", i.e. those that are not positively nor negatively selected), suggesting that the effect of p53 does not increase reproducible variance.

Reviewer #2 (Remarks to the Author):

The authors have addressed most of my concerns. The structure and of the paper is much better now, and it is easier to read. I appreciate that the authors have made a considerable effort to address the points that were raised, and in most cases they have

done so successfully. Of particular value is the new Supplementary Table 7, which reports the p53-related DSB-toxicity scores for several widely-used CRISPR screening libraries. I therefore support the publication of this paper in Nat Commun.

I have a couple of remaining minor comments:

1) Response to point #1, the authors dismiss the potential effect of lentiviral transduction and Cas9 expression on the p53-dependent sgRNA toxicity. I am not convinced that these factors are irrelevant – both the effect of lentiviruses and that of Cas9 expression are likely to preferentially affect specific genomic loci, which may differentially affect different sgRNAs. I think the Discussion paragraph that deals with this topic should acknowledge this.

We have extended the corresponding paragraph in the Results (“Discussion” subsection), where we acknowledge a potential bias that arises from p53 influence on lentiviral transduction and Cas9 expression: *“Incidentally, it has been shown that TP53 status may influence the expression of Cas9¹³, potentially constituting a confounder in our analyses. However, an underexpression of Cas9 in TP53wt cells¹³ should equally affect the efficiency of all sgRNAs in the library, i.e. it would not explain why only one or two of the sgRNAs targeting a gene has different effects between TP53 wild-type and mutant status. Likewise, lentiviral transduction efficiency could be hampered by p53 activity⁵⁸, however, since all sgRNA plasmids are transduced via the same type of lentivirus, again all sgRNAs in the library should be equally affected. Finally, we also acknowledge that lentiviral integration has a preference towards active chromatin⁵⁹, however plausibly the integration site for a particular lentivirus DNA would not be correlated with the sgRNA sequence encoded within (and thus also the sgRNA target site) and so would not confound our analyses”.*

2) Response to point #7: The Chip-Seq analysis of XRCC4 is very nice. The association with genomic features, however, is indeed consistent with the hypothesis but is very circumstantial and does not provide any direct support a role for NHEJ in the described phenomenon.

We agree with the Reviewer that the XRCC4 Chip-Seq analysis is the most direct evidence of the NHEJ hypothesis from point #7. Therefore, we have added the following to Supplementary Text 2D: “*The positive association of XRCC4 Chip-Seq signal with p53-toxic sgRNA targets provides direct evidence for NHEJ involvement, while the association with genomic features such as H3K4me3 is further consistent with it (although alone constitutes circumstantial evidence)*”.

3) *The paper by Sinha et al. (Nat Commun 2021) should be mentioned/cited in the context of the comparison of CRISPR screens between TP53+/+ and TP53-/- cells.*

We would like to thank the reviewer for highlighting this relevant article. The finding by Sinha et al. the sgRNAs of genes that are more essential in TP53wt (vs. KO) cells when using CRISPR-KO (but not shRNA) tend to target accessible chromatin is indeed mirrored in our study, which uses different datasets and methodologies. Therefore, we have cited it at several points of the main text: “[...] *another promising application of CRISPR, in vivo or ex vivo gene editing, whose potential for selection of TP53-/- cells is of concern*^{13,20,22} [...]”, “*The generation and use of TP53-isogenic cell lines has been done before, but for different cell lines: RPE1*^{14,17-19} *and MOLM13*²⁰”, “[...] *Cas9 activity in human cells, when used ex vivo or in vivo for therapeutic purposes, might select for TP53-mutant cells thereby having tumorigenic potential*^{13,20,22} [...]”, “*Overall, the results support that p53 enhances Cas9 DSB toxicity in active chromatin in human cells, in agreement with a recent study*²⁰”, and “[...] *DSBs at sgRNA target sequences that are located in active, accessible chromatin trigger a stronger p53-toxic response, in agreement with a recent study*²⁰”.